

# Moving beyond the cost-loss ratio: Economic assessment of streamflow forecasts for a risk-averse decision maker

Simon Matte[1], Marie-Amélie Boucher[1], Vincent Boucher[2], and Thomas-Charles Fortier Filion[3]

[1]Dept. of Applied Sciences, Université du Québec à Chicoutimi, 555, boulevard de l'Université, Chicoutimi, G7H 2B1, Canada
[2]Dept. of Economics, Université Laval, 1025, avenue des Sciences-Humaines, Québec, G1V 0A6, Canada
[3]Québec Government Direction of Hydrologic Expertise, 675, boul. René Lévesque Est., Québec, G1R 5V7, Canada

*Correspondence to:* Marie-Amélie Boucher (marie-amelie_boucher@uqac.ca)

**Abstract.**

A large effort has been made over the past 10 years to promote the operational use of probabilistic or ensemble streamflow forecasts. Numerous studies have shown that ensemble forecasts are of higher quality than deterministic ones. Many studies also conclude that decisions based on ensemble rather than deterministic forecasts lead to better decisions in the context of

flood mitigation. Hence, it is believed that ensemble forecasts possess a greater economic and social value for both decision makers and the general population. However, the vast majority, if not all, of existing hydro-economic studies rely on a cost-loss ratio framework that assumes a risk-neutral decision maker. To overcome this important flaw, this study borrows from economics and evaluates the economic value of early warning flood systems using the well-known CARA utility function, which explicitly accounts for the level of risk aversion of the decision maker. This new framework allows for the full ex-

ploitation of the information related to a forecasts' uncertainty, making it especially suited for the economic assessment of ensemble or probabilistic forecasts. Rather than comparing deterministic and ensemble forecasts, this study focuses rather on comparing different types of ensemble forecasts. There are multiple ways of assessing and representing forecast uncertainty. Consequently, there exists many different means of building an ensemble forecasting system for future streamflow. One such possibility is to dress deterministic forecasts using the statistics of past error forecasts. Such dressing methods are popular

among operational agencies because of their simplicity and intuitiveness. Another approach is the use of ensemble meteorological forecasts for precipitation and temperature, which are then provided as inputs to one or many hydrological model(s). In this study, three concurrent ensemble streamflow forecasting systems are compared: simple statistically dressed deterministic forecasts, forecasts based on meteorological ensembles and a variant of the latter that also includes an estimation of variable uncertainty. This comparison takes place for the Montmorency River, a small flood-prone watershed in south central Quebec,

Canada. The assessment of forecasts is performed for lead times of one to five days, both in terms of forecasts' quality (relative to the corresponding record of observations) and in terms of economic value, using the new proposed framework based on the CARA utility function. It is found that the economic value of a forecast for a risk-averse decision maker is closely linked to the forecast reliability in predicting the upper tail of the streamflow distribution.



## 1 Introduction

More than fifteen years after its advocation by Krzysztofowicz (2001) and more than a decade after the creation of the HEPEX community (Franz and Ajami, 2005; Schaake et al., 2007), *the case for probabilistic forecasting in hydrology* has been accepted by many researchers and practitioners across the world: uncertainty in hydrological forecasting conveys important information

for decision makers and therefore should be quantified and be considered as part of the forecast. However, as there exists multiple sources of uncertainty in hydrological processes, there also exists many means of assessing this uncertainty and building an ensemble to represent this uncertainty. It is common for operational agencies to resort to analog forecasts (e.g. Hamill and Whitaker, 2006; Diomede et al., 2008; Marty et al., 2012). There are many variants of analog forecasting systems, but all rely on an assessment of past forecasting errors to build the ensemble. Deterministic forecasts can also be "dressed"

using past error statistics. It is also possible to produce streamflow ensemble forecasts from meteorological ensemble forecasts used as inputs to at least one previously calibrated hydrological model. Additional sources of uncertainty can be accounted for, in particular state variable uncertainty or parameter uncertainty. While there is a general agreement among the global scientific community that ensemble forecasts are superior to deterministic ones (e.g. Jaun et al., 2008; Velazquez et al., 2010; He et al., 2013, and many others), there remains no consensus regarding the best means of obtaining an ensemble of streamflow forecasts

(i.e. constructing the ensemble).

There has been increased interest over the last few years in regards to assessing the economic *value* of forecasts. Although the quality of a forecasting system can be assessed by comparing forecasts for different lead times with corresponding observations, forecasts *value* depend on the specific applications and its assessment is not always straightforward (Katz and Murphy, 1997). In particular, the usefulness of a forecast is inherently linked to the decision maker's ability to adapt their behaviour to the

information provided.

In the case of hydropower production, forecast values can be assessed using sophisticated decision-making models based on stochastic dynamic programming in an operational research framework (e.g. Boucher et al., 2012; Carpentier et al., 2013; Côte and Leconte, 2016). Early flood warning is another very important application for streamflow forecasts and a decision problem entirely different from the optimization of hydropower production. Hydrologists most often, if not always, assess the

value of streamflow forecasts for early flood warning using the cost-loss framework (e.g. Murphy, 1977; Richardson, 2000; Roulin, 2007; Verkade and Werner, 2011), which does not fully exploit the information about forecast uncertainty conveyed by the predictive distribution. In particular, it does not account for the decision maker's *risk aversion*, i.e. the fact that, given the opportunity, a decision maker would be willing to spend money (or resources) to reduce the amount of uncertainty they face. This is discussed formally in section 2 below.

This study considers the evaluation of the economic value of early warning flood systems, from the point of view of the decision maker, with explicit consideration of risk aversion. This alternative framework is based on the use of the von Newman and Morgenstern (vNM) utility function (von Neumann and Morgenstern, 1944), which is widely used in economics but rarely in hydrology. To the best of our knowledge, our study represents the first attempt at accounting for risk aversion in the assessment of the economic value of streamflow forecasts for early flood warning. This new framework is used to assess the





economic value of three concurrent streamflow ensemble forecasting systems in a case study for the Montmorency River, a flood-prone watershed in south central Quebec, Canada. Five day statistically dressed deterministic forecasts forecasts for this watershed have been issued operationally since 2008 by the Direction de l'Expertise Hydrique (DEH), a Quebec provincial agency. These forecasts are used for early flood warning and emergency response by the civil security bureau of Quebec City.

It is found that for risk-averse decision makers, the dressed deterministic forecasts have a higher economic value since they provide the most accurate prediction of the upper tail of the distribution.

In section 2, some concerns regarding the cost-loss ratio are raised and an alternative framework is presented. Section 3 describes the context of the case study, namely the specifics of the Montmorency River watershed, the current flood forecasting system based on dressed deterministic forecasts as well as the early flood warning mechanism in place. Two variants of a

concurrent flood forecasting system are detailed in section 3.3. The economic model is presented in section 4. Performance assessment metrics, both in terms of forecasts quality compared to observations, and in terms of economic value, are presented in section 5. Results are presented and discussed in section 6. Conclusions are drawn in section 8 along with suggestions for future improvement of the proposed economic model.

## 2   The economic model and the limits of the cost-loss ratio

The cost-loss ratio decision model (Murphy, 1977; Katz and Murphy, 1997; Richardson, 2000) is a simplified framework used in numerous hydro-meteorological studies to assess the economic value of forecasts (Roulin, 2007; Abaza et al., 2014; Verkade and Werner, 2011,  among many others). As pointed out by Zhu et al. (2002), this approach is only the simplest one out of a much larger range of options. Most importantly, a classical cost-loss ratio decision model disregards the role of risk aversion. In a simple cost-loss ratio, the decision model follows a contingency table that allows for binary decisions, with known associated

costs. When applied to ensemble forecasts, decision-making according to the cost-loss ratio framework is based solely on a probability threshold associated to the material consequences of the event of interest (e.g. a flood event), regardless of the ensemble spread (uncertainty). Appendix A illustrates a technical presentation. Including the concept of risk aversion in the decision model is not only more realistic, but allows weighing the ensemble members differently depending on the level of risk aversion. For instance, a risk-averse decision maker will give more importance to the forecasts members in the upper tail of the

predictive distribution (i.e. highest streamflow values).

The next section describes the formal decision-making process.

### 2.1   A simple economic model with explicit consideration for risk aversion

In economics, "utility" is an ordinal notion that reflects the decision maker's preferences over a set of possible outcomes. In the context of random outcomes, the most popular class of utility functions is the von Newman and Morgenstern (vNM) utility

function, as introduced in (von Neumann and Morgenstern, 1944). The interested reader is referred to Chapter 6 in Mas-Colell et al. (1995) for more details as well as the axiomatic foundations of vNM utility functions.





The vNM utility function of a decision maker regarding a real-valued random outcome $\tilde{c}$ (e.g. money) is given by:

$$U(\tilde{c}) = \sum_{m=1}^{M} p_m \mu(c_m)$$

where $m = 1, ..., M$ are the different "states of the world", $p_m$ is the probability of state $m$, and $c_m$ is the realization of the random outcome $\tilde{c}$ in state $m$. The function $\mu(\cdot)$ is assumed to be non-decreasing.

The curvature of the function $\mu(\cdot)$ reflects the decision maker's preference regarding uncertainty. If $\mu(\cdot)$ is concave, the decision maker is risk-averse; if it is linear, the decision maker is risk-neutral; if it is convex, the decision maker is risk-seeking. Figure 1 displays typical utility curves for risk-averse, risk-neutral and risk-seeking individuals.

This study focuses on a well-known parametric family for $\mu(\cdot)$ known as the Constant Absolute Risk Aversion (CARA) function, given by Eq. 1.

$$\mu(c) = \frac{-\exp(-Ac)}{A} \tag{1}$$

where $A$ is the risk aversion of the decision maker. $A$ is strictly positive for risk-averse individuals, strictly negative for risk-seeking individuals. See Appendix B for additional details.

The parametric form in Eq. 1 implies that the level of risk aversion is independent of the decision maker's financial capacities (hence the name *Constant Absolute* Risk Aversion, CARA). This particular utility function is therefore coherent with the

expected behaviour of most public utility services (municipal authorities will not, for instance, gradually adopt a risk-seeking behaviour regarding the protection of citizens if the city's financial well-being improves).

## 3 Context

### 3.1 Floods on the Montmorency Watershed

Located in southern Québec, Canada, the Montmorency River watershed covers 1150 km$^2$, most part of which is densely

forested. Approximately 30 000 people reside in teh basin, concentrated in its southernmost portion. The northern portion of the watershed lays within the Laurentian Wildlife Reserve, where heavy snowfall precipitation is common. Table 3 presents the seasonal climatological averages for this watershed.

Crystalline rock of the Canadian Shield covers most of the watershed, where the retreat of glaciers left till of an average thickness of 1 m. The southernmost part is covered in sandy sediments from the Champlain Sea. Figure 2 shows the geographi-

cal location of the watershed as well as the location of the available meteorological stations and streamflow gauges (see section 3.3).

The Montmorency River experiences quasi-annual ice jams during spring melt, which often enhance the magnitude and frequency of floods within vulnerable inhabited areas. The response time of the watershed is rapid (12 hours). The return period of damaging floods is also short. This makes emergency evacuation and flood damage a common occurrence for riverside

residents. Table 1 shows return periods and corresponding streamflow values for the Montmorency River (Leclerc and Secretan,





2012). Note that these are given for open-water levels, and take neither ice jams nor the increase in water level due to the presence of ice blocks into account.

The behaviour and consequences of ice jams along the Montmorency River have been the focus of previous studies, such as forecasting river ice breakup (Turcotte and Morse, 2015). Risk analysis and technical solutions (Leclerc et al., 2001) have also

been studied, but as of yet not implemented.

The river experienced its worst recorded event in 1964, when a heavy rain system melted a late autumn snow cover, resulting in a 1100 m$^3$/s flow peak. More recently, an ice cover breakup followed by the formation of an ice jam formation further downstream in January 2008 forced the evacuation of 80 households and damaged four houses. In March 2012, an early spring thaw caused by extreme temperatures caused a flood resulting in the evacuation of 25 households. More recently, in April 2014,

an ice jam breakup caused a massive ice-carrying flood wave that, occurring during a typical normal spring freshet, quickly raised waters to a semi-centennial level. In addition, the topography in the area causes certain regions to become entirely isolated and surrounded by water during flooding. The greatest concern of public authorities occurs when people refuse to evacuate, especially in these flood-prone areas.

### 3.2   Current forecasting and decision-making process

### 3.2.1   The hydrological model HYDROTEL

HYDROTEL (Fortin et al., 1995) is a spatially distributed, physics-based model developed and maintained by the Institut National de Recherche Scientifique (INRS). It is used operationally by the DEH, and has been implemented in the Montmorency River watershed since 2008 (Rousseau et al., 2008). The model accepts gridded inputs (precipitation, snow cover, temperature) than can be interpolated using a three station average or Thiessen method. Physical features of the catchment (topography, soil

type, hydrographic network) are processed by a companion software called PHYSITEL. It divides the watershed in smaller spatial units called RHHU (Relatively Homogeneous Hydrological Units). Each of the RHHU is then assumed to possess homogeneous physical properties. HYDROTEL then performs the computation of vertical and horizontal water flows.

HYDROTEL offers a range of sub-routines for hydrological processes (interpolation of precipitation, evapotranspiration, snow accumulation and melt, etc.). The user chooses the most appropriate sub-routines depending on the available data. For

this study, interpolation of observed precipitation was performed using Thiessen's polygons. No radiation data were available, so evapotranspiration was estimated from an empirical temperature-base method (Fortin, 2000; Bisson and Roberge, 1983) and snowmelt was modelled by a mixed degree-day/energy budget approach. The vertical water budget was performed by the sub-routine BV3C that divides the soil into three layers of different composition and depths. Both overland and channel routing was performed using the kinematic wave approach (Lighthill and Whitham, 1955). With this setup, which replicates the model

setup used operationally by the DEH, HYDROTEL has 27 parameters, but only 10 were calibrated (default values were used for the other parameters). The calibration already performed by the DEH was kept intact. In forecasting mode, HYDROTEL is driven by meteorological forecasts, either deterministic or ensemble-based. At each time step of a simulation, HYDROTEL updates and save state variables into four separated files: one for surface runoff, one for snow, one for soil moisture and one for





streamflow. For instance, the snow file comprises values for five spatially distributed snow-related state variables: snow water equivalent (SWE), snow cover depth, calorific deficit, liquid water content and albedo. In the actual operational setting, data assimilation is performed manually and indirectly: the forecaster modifies precipitation and/or temperature observed during the previous days until the model's simulation is in agreement with the observed streamflow for the actual day. When the model

is run with the modified meteorological inputs, state variables are re-computed and should translate into an improvement of the model's description of the hydrological state of the watershed. The choice of applying modifications to temperature or to precipitation depends mostly on the period of the year and associated dominant hydrological that is processed. Thus, during spring freshet, air temperature is the main forcing that acts on the snow melt rate. Solar radiation is not among HYDROTEL's inputs but is rather estimated empirically, in part through air temperature. Therefore, during this period of year (early March

to late May), perturbations are applied on temperature forcing. During the summer and early fall periods, precipitation forcing is the dominant factor for controlling runoff, soil moisture and eventually streamflow. Perturbations are applied primarily on precipitation from approximately June to November.

### 3.2.2 Flood alerts

The Direction de l'Expertise Hydrique (DEH) is an administrative unit of the Government of Québec created in 2001 with

the mandate to manage the water regime of Québec's rivers and provide streamflow forecasts to municipalities. Since 2008, operational five day, three hour time step streamflow forecasts are distributed to municipal water managers. Those forecasts are always obtained using the semi-distributed physics-based hydrologic model HYDROTEL (Fortin et al., 1995). Although HYDROTEL is a deterministic model, the operational forecasts now largely distributed by the DEH are not purely deterministic but are rather accompanied by a 50% confidence interval. This confidence interval is computed from a statistical model derived

from the analysis of past deterministic streamflow forecasts errors for 10 watersheds across the province of Québec. A more detailed description of this statistical method is available in Huard (2013).

After receiving a forecast exceeding a pre-determined flood threshold, municipalities can choose to engage in emergency procedures. In the case of the Montmorency watershed, current measures are mostly reactive (road closure, controlled evacuation of citizens, providing emergency shelters and food) rather than preventive (artificial levees, culverts, etc., Leclerc et al.,

25 2001).

Flood thresholds have been adapted from an hydrodynamic study (Leclerc and Secretan, 2012). Threshold numbers have been conservatively rounded down to compensate for the worsening effect of ice in the channel. Table 2 shows operational threshold levels for the most vulnerable residential area.

### 3.3 A concurrent flood forecasting framework based on meteorological ensemble forecasts

The alternative forecasting framework proposed in this study involves meteorological ensemble forecasts passed on to HY-DROTEL. Those forecasts were obtained from the Thorpex Interactive Great Grand Ensemble (TIGGE) database managed by the European Center for Medium Range Weather Forecasts (ECMWF). Precipitation and temperature ensemble forecasts from the Meteorological Service of Canada (MSC) were retrieved, covering the 2011—2014 period . The forecasting horizon is five





days, with a six hour time step. The MSC meteorological ensemble forecasts comprise 20 members. The initial spatial grid of $0.6°$ was downscaled to a $0.1°$ grid through simple bi-linear interpolation during data retrieval.

Observations for precipitation and temperature are measured at five ground stations distributed around the watershed (see Figure 2). Hourly measured data was accumulated and averaged over a three hour time step. Snow survey data interpolated on a $0.1°$ grid are also available. They were provided for this study by the DEH. The streamflow gauging station at the river outlet provides measurements at a 15 minute interval, corrected for backwater due to ice cover and then averaged over three hour time steps.

### 3.3.1 Data assimilation and state variable uncertainty

Appropriate data assimilation is crucial for short-term flood forecasting as it allows the model to begin the forecasting period having the best possible estimate for initial conditions. Thiboult et al. (2016) showed that the uncertainty for initial conditions dominates the other sources of uncertainty for short-term (1-day to 3-day ahead) streamflow forecasts.

In this study, manual data assimilation was performed according to the guidelines by Mamono (2010) and agree with the procedure followed by operational forecasters at the DEH. This assimilation process relies on the assumptions that: (1) model errors are entirely compensated by the model calibration process, (2) streamflow measurements are error-free, and (3) the only remaining source of error affecting state variables is attributable to meteorological inputs (Mamono, 2010). Additive coefficients between -10 and 10 °C were applied to temperature inputs while multiplicative coefficients between 0.1 and 10 were applied to precipitation inputs in order to improve the agreement between simulated and observed streamflow series. Those perturbations of meteorological inputs were applied uniformly onto the basin for fixed periods of time.

The manual data assimilation described above only improves on the "best guess" of the state variables for each time step. To go one step further, additional perturbations were applied around this best guess estimate in order to account for the uncertainty of initial conditions. To do so, a rudimentary Ensemble Kalman Filter (EnKF, Evensen, 2003) was implemented. From the starting point—constituted by manually assimilated precipitation, temperature and streamflow simulation series—random noise is further applied to precipitation and temperature inputs. Additive perturbations are drawn randomly from $U(-8,8)°$ for temperature. For precipitation, both multiplicative ($U(0.5, 1.5)$) and additive ($U(0, 0.5)$ mm) perturbations are drawn. The inclusion of additive perturbations for precipitation is due to the fact that strong under-captation is suspected for this catchment. Output uncertainty is modelled by a normal distribution centered on observed streamflow with a standard deviation taken as 10% of the observed streamflow. In this study, data assimilation is a necessity rather than a choice and is not at all the primary objective. For this reason, the limits of the above-mentioned distributions were not optimized as in Thiboult and Anctil (2015). Those limits were fixed according to the guidelines in Mamono (2010) and Abaza et al. (2015) and the experience gained during manual data assimilation. Further refinements of the EnKF model is outside the scope of this study.

The Kalman gain $K$ is computed following Mandel (2006)

$$\boldsymbol{K}_t = \boldsymbol{M}_t \boldsymbol{H}^T (\boldsymbol{H} \boldsymbol{M}_t \boldsymbol{H}_t^T + \boldsymbol{O}_t)^{-1} \tag{2}$$





where $\boldsymbol{M}_t$ is the model error covariance matrix computed according to the perturbations defined above and $\boldsymbol{O}_t$ is the covariance of observation noise also computed according to the perturbations drawn from the normal distribution described above. The matrix $\boldsymbol{H}$ relates the state vectors and observations (so called "observation model"). It can be demonstrated through matrix algebra that Eq. 2 amounts to computing the derivative of the analysis error and setting it equal to zero.

Once the Kalman gain is computed, it is used to weight the credibility of the model error $z_t - \boldsymbol{H}\boldsymbol{X}^-$ relative to the a priori estimation of state variables $\boldsymbol{X}^-$ according to Eq. 3. This leads to the updated model states, $\boldsymbol{X}^+$.

$$\boldsymbol{X}_t^+ = \boldsymbol{X}_t^- + \boldsymbol{K}_t(z_t - \boldsymbol{H}\boldsymbol{X}_t^-) \tag{3}$$

The next section adapts the general framework presented in section 2.1 to the specifics of the Montmorency watershed.

## 4    Parametrization of the economic model

The preferences of a decision maker with risk-averse preferences represented by a CARA utility function can be represented as follows:

$$U(s) = \sum_m p_m \frac{-1}{A} \exp\left\{ -A\Big[ -d(Q_m) + b\big(d(Q_m), s, w\big) - s \Big] \right\} \tag{4}$$

Strictly speaking, each ensemble member $Q_m$ has a probability of occurrence $p_m$, and corresponds to a given damage $d(Q_m)$. In this study, the damage curve is broken down into 12 categories (i.e. $m = 1, ..., 12$). Then, $Q_m$ represents the stream-

flow associated with the $m^{th}$ category and $p_m$ becomes the probability associated to this category, inferred from the number of members that fall within it. Given $s$, the amount of money spent ($w$ days ahead, see section 4.3 below) on flood emergency measures, the resulting gain (or benefit) in terms of damage reduction is given by $b\big(d(Q_m), s, w\big)$.

While $Q_m$ and $p_m$ are derived directly from the ensemble forecast, $d$, $s$ and $b\big(d(Q_m), s, w\big)$ must be calibrated from other sources of information related to actual operation and decision history. This can be a challenge, but fortunately in the case of

the Montmorency River, a record of citizen evacuations and corresponding spending for the 2014 flood was available. Although incomplete, this record allows us to guide the estimation of $d$, $s$ and $b\big(d(Q_m), s, w\big)$.

### 4.1    Level of risk aversion $A$

Risk aversion $A$ is an intrinsic characteristic of each person or organization and could be calculated, given the availability of a sufficiently long record of decisions and associated money spending. However, in the present study, $A$ was left free for the

following reasons. First, the available data is not sufficient to credibly calibrate $A$. Second, as one of the goals of this study is to illustrate how risk aversion influences the value of a forecasting system for a particular problem, it is logical to cover a range of possible $A$s, including the risk-neutral $A = 0$ situation. Therefore, $A$ was made to vary from 0 to 0.01. Although these represent relatively small levels of risk aversion, it is shown that they lead to qualitative changes in the decision maker's





spending decisions, and therefore on the economic value of the concurrent forecasting frameworks. Negative values for $A$ were not considered, as they represent a risk-seeking decision maker, unrealistic in the context of flood mitigation.

## 4.2 Damages $d$, spending $s$ and damage reduction $b$

The material damages to houses and property associated with flood events can be estimated using the flow-damage curve
established by Leclerc et al. (2001). The damage expressed in dollars rises exponentially with observed streamflow ($m^3/s$) and range from \$0 to \$375 000.

In this study, the following parametrization of the benefit function is used:

$$b\big(d(Q_m), s, w\big) = \min\big\{\beta_w \cdot s, \psi \cdot \hat{d}(m)\big\}$$

where $d(m) = \psi\hat{d}(m)$, $\hat{d}(m)$ is the flow-damage curve (Leclerc et al., 2001) for the forecast member $m$, and $\beta_w$ and $\psi$ are
parameters. This particular parametrization assumes that the benefit of spending is linear, until all damages are avoided. It also implies that it is never optimal to spend more than $\max_m\{\psi \cdot \hat{d}(m)\}$, since additional spending brings no additional benefit, for any possible forecast member.

The parameter $\beta_w$ has been initially calibrated by assuming $\psi = 1$. By comparing the total amount of money spent in 2014 to alleviate flood damages with the damages (in dollars) predicted by the aforementioned damage curve using the observed
streamflow, it was found that the calibrated $\beta_w$ was less than one. This implies that the civil security service would have spend more than the total amount of possible damage.

This therefore implies the existence of intangible benefits associated with having a flood warning system and spending money to mitigate flood effects. According to Lave and Lave (1991) and Carsell et al. (2004), these intangible benefits include but are not limited to: not putting people's health and security at risk, stress reduction for the population, and building a feeling of trust
towards the authorities. In the case of the Montmorency River, there has never been any loss of life. However, as mentioned earlier, it may happen that people refuse to leave their residences and become isolated from communicating roads restricting their access to services and medical care. Unfortunately, it is very difficult and probably rather imprudent to associate a definite cost to these intangible benefits such as "reducing stress". In the absence of a better alternative, in this study a multiplying factor $\psi$ was applied to the damage curve to account for those intangible benefits, as suggested in Van Dantzig and Kriens
(1960). The parameter $\psi$ was made to vary between 1.5 to 10 and $\beta_w$ was computed again for each different value of $\psi$, as the damage curve is modified. The lower limit of $\psi$ was set so that money spent during the flood of 2014 equals the damage predicted by the damage curve. Therefore, in this framework, the damage curve of Leclerc et al. (2001) (i.e. $\hat{d}(m)$) represents mostly the relationship between streamflow and its impact on the lives and well-being of people.

## 4.3 Warning time and dynamic decision-making

According to USACE (1994), Richardson (2000) and Roulin (2007), the costs of emergency measures or benefits thereof are related to warning time $w$. In particular, Roulin (2007) assumes that early action can reduce the total cost of emergency





measures and maximize damage reduction. Carsell et al. (2004) also provide an evaluation of residential content (furniture, food, electric appliances, etc.) that can be protected with a given warning time.

That being said, the decision-making process analyzed in this study is a static one. If the analysis of a dynamic decision process would be more realistic, it would also introduce many more questions regarding how the total spending should be distributed among lead times. For instance, depending on their level of confidence regarding the 5-day forecast, a decision maker could decide to launch an evacuation alert and immediately spend all available funds for emergency measures. As stated in Roulin (2007), intuition lends to thinking that preparing in advance for a flood could lead to reduced overall spending compared with waiting until the last minute. This is also discussed in Morss (2010) in her analysis of three case studies of the interactions between flood forecasts, decisions and outcomes. She provides examples of the importance of early actions:

*Key property- and life-saving decisions are often thought of as taking specific protective action immediately prior to or during an event. However, sometimes key decisions can be less evident and occur during earlier planning stages. For example, in Grand Forks, once officials had decided to expend most of their time, effort, and resources on planning and building primary dikes, they were not able to plan and build secondary dikes fast enough when the flood grew worse than expected. In the Pescadero case, if officials had not decided to position rescue crews and equipment before the flood began, they would not have been able to reach the area.*

However, the accuracy of forecasts is inversely proportional to lead time and the decision maker might want to wait for better information before taking a decision.

Those considerations go far beyond the objective of this study, and the formalization of an explicit dynamic decision process is left for further research. In this study, the dynamic nature of the problem is addressed by assuming that the decision maker uses the following myopic decision procedure:

1. At the beginning of each day, the decision maker receives a 5-day forecast.

2. Iteratively, and starting with the *earliest* (5-day) forecast, the decision maker chooses their preferred level of spending. This level of spending is chosen as to maximize Eq. 4.

3. The decision maker is constrained (by external factors such as the availability of materials or labour force) to spend at most a fraction $\delta$ of their preferred level of spending $s$ (see below).

The benefits of a spending are assumed to take effect on the day the spending decision is made, up until the forecast date. For example, if decision maker spends \$1000 on a given Monday, anticipating a flood the following Thursday (i.e. a 4-day forecast), then any damage occurring prior to Thursday is also reduced (by $\beta_w \times \$1000$).

The parameter $\beta_w$ is divided between lead times according to $[2, 1.75, 1.5, 1.25, 1]\beta_{2014}$, where $\beta_{2014}$ is calibrated on the spending decisions of 2014 and represents the baseline ratio of gain per dollar invested. The above multiplication therefore assumes that early actions lead to higher gains per dollar spent. This is very similar to the methodology presented in section 4.3 of Roulin (2007), except that only one repartition of $\beta_w$ is tested here compared to two in Roulin (2007).

If the decision maker is to take successive actions at different lead times according to forecasted streamflow, then the total amount of available money can be spread across lead times. The decision maker can, for instance, spend all the available money





two days prior to the event. Or, they can spend half two days prior and the remaining half the day before the flood (1-day). To account for this, five different "spending vectors" were created (Table 4). The values in those spending vectors represent the maximal fraction $\delta$ of the preferred level of spending $s$ that can be spent at each lead time. The first three spending vectors represent situations for which there is no limit on the spending than can be made the day before, with spending vector number 3

representing the extreme case where the decision maker *must* wait until the 1-day forecast before spending any money. On the contrary, spending vectors number 4 and 5 represent a fictitious situation in which the decision maker can spend any amount of money at the 5-day horizon, and no spending is allowed the day before (1-day).

It is important to note that due to the myopic decision-making procedure, the decision maker does not take into account the fact that money spreads across lead times when making a decision. This effect alone underestimates the value of early spending.

However, the decision maker also does not consider the reduction in uncertainty gained by waiting (which overestimates the value of early spending). In this study, those two effects are assumed to balance each other.

## 5 Performance assessment

### 5.1 Forecast quality

The three forecasting systems described in sections 3.2 and 3.3 are compared to each other by assessing their respective abilities

to forecast observed streamflow values for the 1- to 5-day projections. This performance assessment involves the well-known Continuous Ranked Probability Score (CRPS, Matheson and Winkler, 1976) and a reliability diagram (Stanski et al., 1989). A visual inspection of corresponding hydrographs is then undertaken.

### 5.2 Evaluating the benefits of forecasts

As described in the introduction, the usefulness of an early flood warning system is in helping the decision maker choose the

best spending level $s$, prior to the event. The value of such system is therefore closely related to the decision maker's ability to affect the outcome through their spending decisions. The benefits of forecasts are therefore evaluated with an explicit concern for the decision maker's preferences.

In order to develop an indicator of the economic benefits of a forecast, it is important to distinguish between the decision maker's *ex-ante* utility (before the uncertainty is resolved, as in Eq. 4) and their *ex-post* utility (the realized level of utility, after

the uncertainty is resolved). This is important as spending decisions are based on the *ex-ante* utility, whereas the value of the forecasts are based on the (expected) *ex-post* utility, conditional to spending decisions. Given the spending decision $s$ and the realized state $m$, the *ex-post* utility of the decision maker is given by:

$$U_m(s_f) = \frac{-1}{A} \exp\left\{-A\left[-d(Q_m) + b\big(d(Q_m, s_f, w) - s_f\big)\right]\right\} \tag{5}$$

where $s_f$ is the total amount of money spent, from a decision based on forecasts (f). The value of this *ex-post* utility is dependent, of course, of the realized streamflow values. In order to obtain a sensible evaluation of the decision maker's utility,





one must therefore consider the average *ex-post* utility:

$$\mathbb{E}_m U_m(s)$$

where the expectation $\mathbb{E}_m$ is taken with respect to the historical streamflow values. Note that the history under consideration must be long enough to be representative of the true distribution of streamflow.

Utility can be computed for any of the three forecasting systems described in sections 3.2.2 and 3.3 but also for two special cases: perfect forecasts and no forecasts. On one hand, if forecasts were perfect, there would be no missed events and the

decision maker would spend only the exact amount of money necessary to obtain the maximum possible protection, as early as time allowed. On the other hand, if no forecasts were available, there would be no decisions to be made and no money to be spent on flood mitigation and protection measures. Therefore, the maximum amount of damage would occur for each flood event.

It is important to note that utility is an ordinal quantity that only represents the preference of a person faced with a decision-

making problem, given some information from uncertain forecasts. That is, the utility levels can be compared, but the actual value of the decision maker's utility has no interpretation. Consequently, the utility values computed for the three forecasting systems can be scaled relative to the utility of a perfect forecasting system. This simplifies the interpretation, without imposing any additional restriction.

The hit rate and the overspending index, two standard measures of the economic performance are also presented.

The hit rate, given by Eq. 6, is the ratio of avoided damages when decision-making is based on the forecasting system being evaluated to the damages that would be avoided if the forecasts were perfect (always equal to the observations).

$$Hit\ Rate = \frac{\mathbb{E}_m b\big(d(Q_m), s_f, w\big)}{\mathbb{E}_m b\big(d(Q_m), s_p, w\big)} \tag{6}$$

where $s_p$ is the amount of money that would have been spent if perfect forecasts would have been available. $s_f$ is the total amount of money spent when decisions are based on forecasts, as in Eq. 5. $s_p$ matches exactly the damages corresponding to

the observed streamflow, for all time steps.

Overspending is defined as in Eq. 7. It allows for measuring how much the forecasting system being evaluated overspends (in percentage) compared to perfect forecasts. One should aim for the overspending value to be as low as possible.

$$Overspending = \frac{\mathbb{E}_m s_f - \mathbb{E}_m s_p}{\mathbb{E}_m s_p} \tag{7}$$

Results are presented in the next section. A brief description of the simulation procedure for the computation of the economic

value of forecasts can be found in Appendix C.





## 6 Results

### 6.1 Assessment hydrological forecasts relative to observations

Figure 3 displays hydrographs for a two-week period during the spring of 2014. Panels (a), (c) and (e) correspond to 1-day forecasts while panels (b), (d) and (f) correspond to 5-day forecasts. In all cases the time step is three hours. Forecasts along the upper row (a and b) are dressed deterministic forecasts. Forecasts along the middle row are based on meteorological ensemble forecasts without EnKF while forecasts on the bottom row are also based on meteorological forecasts but account for state variables uncertainty through EnKF. This figure shows that for 1-day forecasts, those based on meteorological ensembles and dressed deterministic forecasts have similar spread. This is expected, as only the forcing uncertainty is accounted for and this uncertainty requires more than one day to be propagated through the hydrological model. The influence of the EnKF can also be seen. The spread of the forecasts with EnKF is greater than the forecasts without EnKF and the density of forecasts members is higher around the observed streamflow. At the 5-day lead time, some members of the forecasts based on meteorological ensembles reach very high streamflow values. This is not the case for the dressed deterministic forecasts that often underestimate streamflow.

Figure 4 presents the mean CRPS of the three concurrent forecasting systems over the 2011—2014 period. The CRPS was computed separately for each lead time in three hour increments and averaged over the entire record of forecasts and corresponding observations. For very short lead times, the dressed deterministic forecasts outperform those based on meteorological ensembles (lower CRPS). However, for lead times longer than 18 hours, forecasts based on meteorological ensembles achieve a better (lower) CRPS than dressed forecasts, despite the jumpy behaviour of the ensemble curves compared to that of the dressed forecasts. Furthermore, the performance gap between meteorological ensemble-based forecasts and dressed forecasts increases with lead time.

The perturbation of state variables after manual data assimilation increases (worsens) the CRPS. This is likely attributable to a loss of resolution. Although sharpness and resolution are two desirable attributes of a forecasting system, there is most often a trade-off between the two. Indeed, Figure 5 highlights that forecasts based on meteorological ensembles having a perturbation of state variables display a better reliability than when state variables remain unperturbed. The difference is most striking for 1-day forecasts. Figure 5 also shows that dressed deterministic forecasts are more reliable than forecasts based on meteorological ensembles for short lead times (e.g. one day, hollow circles), but less so for longer lead times (e.g. 5-day, hollow triangles). As lead time increases, the accuracy of meteorological forecasts decreases. However, the spread of forecasts based on meteorological ensembles increases considerably with lead times therefore more often including the observed values at the 5-day lead time compared to the 1-day lead time.

### 6.2 Assessment of hydrological forecasts in terms of economic value

For each of the simulated values of $A$ and $\psi$, the application of each spending vector (c.f. Table 4) was tested over the study period (2011-2014). This section describes the simulation procedure. Additional details are found in Appendix C.





An example of the applied methodology and corresponding results is provided in Figure 6. The upper row shows 5-day forecasts from the three systems, starting on May 17, 2014. The lower row shows how each member of each forecast is classified into 12 severity classes ranging from non-damaging (class 1) to centennial-scale flooding (class 12) defined after the damage curve.

The utility function (eq. 4) is used successively with the five spending vectors presented in Table 4. The probabilities $p_m$ with $m = 1...12$ in Eq. 4 correspond to the relative frequencies of each category after classification of forecast members that allows for computing the utility as a function of the money spent. The utility curve maximum provides the optimal spending associated with each forecast. Figure 7 illustrates an example for $A = 0.01$ and $\psi = 7$.

Figure 8 presents the utility, hit rate and overspending as a function of parameter $\psi$ for the three flood forecasting systems
under study for various levels of risk aversion and for spending vector number 1 (see Table 4). Note that $A = 0$ corresponds to the case of a risk-neutral decision maker. Negative risk aversion values representing risk-seeking behaviour, were not used. As mentioned in section 5.2, any affine transformation of the utility function is admissible. In Figure 8, the utility of a perfect forecast was subtracted from the utility of each concurrent forecasting system and from the "no forecast" situation. This allows the y-axis of the utility plots to start at 0 and provide a common reference. This figure shows that a risk-neutral decision maker
prefers having information from forecasts based on meteorological ensembles (with or without EnKF) rather than having no forecasts. However, for higher levels of risk aversion ($A = 0.01$, bottom line of Figure 8), the decision maker should prefer the "no forecast" situation for low levels of $\psi$.

Although this seems counter-intuitive, it can easily be explained by looking at the hydrographs (cf. Figure 3). Forecasts based on meteorological ensembles, in particular using EnKF, have a tendency to generate members with very high streamflow
levels. As risk aversion increases, the decision maker puts more weight towards those members, as the associated damage is considerable. This causes the decision maker to spend large amounts of money to "insure" against the potential damage.

As such high streamflow levels are historically rare for the Montmorency River, the decision maker would have been better off not to spend any money and suffer damage during the relatively rare and comparatively small flood events.

Dressed deterministic forecasts decrease weakly with $\psi$, relative to the ensemble forecasts. Put differently, for large amounts
of material damage, the dressed deterministic forecasts have much higher values than the ensemble forecasts. This is due to the fact that, for all lead times, ensemble forecasts include members having "unrealistic" streamflow values. This over-forecasting is exacerbated for high values of material damage and a high value of risk aversion. As the concavity of $\mu$ increases (due to an increase in the level of risk aversion $A$), "bad shocks" are weighted more heavily by the decision maker, leading to considerable levels of (over-) spending.

The same effect can be seen for alternative choices of spending vectors. Figure 9 shows the same parameters (utility, hit rate and overspending) as a function of $\psi$, for the same forecasts, but for spending vector number 2. With this spending vector, the decision maker cannot spend any amount of money five days ahead and can then progressively spend a greater percentage of the available money as the lead time decreases. In such a case, the decision maker should prefer to have access to forecasts based on meteorological ensembles (rather than the no forecast situation) if they are slightly risk-averse ($A = 0.001$). This





is explained by the fact that the 5-day forecast (which contains extreme forecast members, c.f. Figure 3) is not used by the decision maker, which limits overspending.

Eventually, a more risk-averse decision maker ($A = 0.01$) should prefer the dressed forecasts over any other forecasting system, for $\psi$ values over 6. This is again attributable mostly to some members of the ensemble systems frequently forecasting

flood events that don't materialize. This is confirmed by the overspending graphs on the right-hand side of Figure 9. Hence, in Eq. 4, the optimal level of spending $s$ is less for the dressed forecasts than for the other forecasting systems.

When $\psi$ becomes very large (very important material damages) the utility of the "no forecast" framework decreases rapidly, especially for a more risk-averse decision maker. Then, even if the decision maker generally overspends, all forecasts are preferred to the "no forecast" situation since the damage associated with a flood event are considerable. For high values of $\psi$,

the spending decision effectively acts as an (valuable) insurance policy. The hit rate increases (slightly) with the level of risk aversion. This is expected, as a risk-averse decision maker will attribute more importance to large streamflow values in the ensemble forecast.

The third column of Figure 9 shows that a risk-averse decision maker would reduce their overspending by using a forecasting system based on dressed deterministic forecasts rather than on meteorological ensemble forecasts with or without EnKF.

Dressed deterministic forecasts exhibit much less dispersion than EnKF forecasts, which also accounts for state variable uncertainty. As it was remarked earlier, a risk-averse decision maker will put more weight on higher streamflow values in the ensemble. If the spread is large, the ensemble necessarily includes larger streamflow values. It is therefore not surprising that overspending is larger for the ensemble forecast with the larger spread, especially for high values of both $A$ and $\psi$.

The results for the other spending vectors (c.f. Table 4) are qualitatively similar and are therefore not presented. These results

are available as supplementary material.

Figure 10 shows boxplots of $Q_{f,max} - Q_{obs}$ for dressed forecasts, forecasts based on meteorological ensembles and forecasts based on meteorological ensembles with state variable uncertainty (EnKF). This graph focuses only on the highest streamflow value of each daily forecast, $Q_{f,max}$. This is the worst-case scenario. It is readily apparent that the worst-case scenario of the dressed forecasts is often lower than the corresponding observed value (i.e. $Q_{f,max} < Q_{obs}$). Dressed deterministic can both

over- and under-predict streamflow, with comparable frequencies. The two other forecasting systems most often over-predict streamflow. In addition, when $Q_{f,max} > Q_{obs}$ for dressed forecasts, the magnitude of exceedance is, in most cases, far less than for the two other systems. This implies that the worst-case scenario according to dressed forecasts is usually not as bad as when using meteorological ensembles to issue streamflow forecasts. Our risk-averse decision maker is then less inclined to overspend. Figure 10 shows that such extremes can be *even greater* for forecasts based on meteorological ensembles. Of

course, the highest values on Figure 10 (b) and (c) are outliers, meaning that they are exceptions to the general rule. However, even a limited number of largely over-forecasted events can trigger useless spending and therefore decrease the economic value of forecasts. In can also be noted that the difference between forecasts based on meteorological ensembles without EnKF (b) and with EnKF (c) lies in the representation of extreme events at the 1-day lead time. There are more such over-forecasted situations at this lead time when the EnKF is used as part of the forecasting system. This is sufficient for the EnKF forecasts to

have lower economic value than the forecasts relying only on meteorological ensembles.





## 7 Discussion

The "real" level of risk aversion for the decision maker for flood emergency measures along the Montmorency River remains unknown due to the insufficient record of decisions and associated spending. However, it can be reasonably assumed that they are highly risk-averse (Claude Pigeon, personal communications). Considering $A = 0.01$ and Figure 9, the dressed determin-
istic forecasts provide maximal utility. They have a lower hit rate but also a much lower level of overspending compared to the other forecasting systems. This leads to the conclusion that dressed forecasts have the highest economic value for this level of risk aversion.

However, this conclusion relies on the assumption that benefits are linear. As the level of damage (i.e. $\psi$) increases, so does the spending needed to alleviate this damage. In a situation where human casualties are possible (resulting in extremely high
values of $\psi$), the spending needs not to increase with $\psi$. For example, the cost of an evacuation is not linked to the (somewhat subjective) value associated with human casualties. These considerations are left for further research.

The economic value of a forecasting system is necessarily dependent on the level of risk aversion of the decision maker. However, the role of the forecaster is to issue the best possible streamflow forecast given their knowledge of the situation and available model and data. It is the end-user's role to decide the course of action. Of course, effective communication and
interactions between the forecaster and the user are also key components of the complete flood awareness and preparation scheme. However, this clear separation of roles is not always as evident in practice. For example in Danhelka (2015):

*The Minister simply asked me what the forecast for Prague was. After I have explained all the known information, forecasts and uncertainties, I gave him my best guess of the peak flow. But his response was: "No, no, no, give me the worst-case scenario; don't tell me numbers you cannot guarantee as not being exceeded".*

Here, the user (the Minister) is expressing that he is (infinitely) risk-averse. He will base his decisions on the worst-case scenario. However, this position does not make the forecast irrelevant as the other ensemble members might still be useful to other decision makers.

To a certain extent, this then becomes a communication problem. Deliberately "over-forecasting" for a particular end-user is, according to the authors, a dangerous practice as it is difficult to draw the line. However, we also agree with the general
consensus that communicating the forecast in an adequate manner, depending on the needs and preferences of specific end-users, remains important.

As illustrated by Figure 10, the worst-case scenario of the forecast ensemble can easily be greater than the corresponding observations. For the above-mentioned government minister, who wishes to base their decisions on this particular scenario, this will necessarily lead to overspending and a low forecast utility. That being said, in the above citation, the minister was referring
to a particular flood event and not to every single forecasts. In addition, most decision makers are not *infinitely* risk-averse and will not only consider the most extreme member of the ensemble. Nevertheless, there is a key conclusion to emphasize here. For a forecasting system to be useful (in an economic sense) to a risk-averse decision maker, extra care must be taken in the design of said system to ensure that the upper tail of the predictive distributions are accurate.





Our study also shows that forecast *quality* (as verified using metrics such as the CRPS) is not always a guarantee of forecast *value* in an economic sense. In this study, the streamflow forecasts based on meteorological ensembles have better CRPS than dressed deterministic forecasts. The addition of state variable uncertainty worsens slightly the CRPS but it increases the reliability of forecasts. However, for a risk-averse decision maker, forecast members in the upper tail of the predictive

distribution are heavily weighted. As dressed deterministic forecasts provide less "extreme" members, they are more valuable for this application of flood mitigation by a risk-averse decision maker.

## 8 Conclusions

The purpose of this study is to set the basis of an alternative framework to replace the cost-loss ratio in economic assessment of early warning flood forecasting systems. This alternative framework is based on the Constant Absolute Risk Aversion (CARA)

utility function which is well-known in economics. To the authors' knowledge, risk aversion is rarely, if ever, accounted for in hydro-economic assessment of flood warning systems. This new framework is used to compare the economic value of three concurrent streamflow ensemble forecasting systems using the flood-prone Montmorency River watershed in Quebec, Canada. This study concentrates on ensemble rather than deterministic forecasts, as the recent literature clearly states that ensemble forecasts are preferable to deterministic ones for numerous reasons (e.g. Krzysztofowicz, 2001; Jaun et al., 2008;

Velazquez et al., 2010; He et al., 2013). Furthermore, real-life operations for the case study involved here (flood forecasting for the Montmorency River) do not involve deterministic forecasts. However, there exists many different means of constructing streamflow ensemble forecasts: dressed deterministic forecasts, single hydrological models fed with meteorological ensemble forecasts, multiple hydrological models, with or without data assimilation, etc. Those different forecasting systems can be compared in terms of their correspondence with the observation an in terms of their value for an end-user.

The importance of the level of risk aversion of the decision maker for the determination of the economic value of a streamflow forecasting system is illustrated by our results. A risk-neutral decision maker, as assumed in the cost-loss ratio framework, is rarely, if ever, encountered in real-life decision problems. The value of forecasting systems strongly depends on the decision maker's level of risk aversion and this parameter should be as much as possible targeted to the end-user. The results also show that forecast quality as assessed by the CRPS, or the reliability diagram, do not necessarily translate directly into a greater

economic value, especially if the decision maker is not risk-neutral. Frequent over-forecasting strongly affects the economic value of forecasts. Over-forecasting can be corrected by adequate statistical post-processing of the predictive distributions. This was judged outside of the scope of this study but could certainly be explored in further work. Adequate post-processing would likely improve the value of forecasts.

The decision-making framework presented here can be improved in some ways. Further studies could also include a more

detailed, dynamic decision-making process, formally accounting for the forecast horizon. Furthermore, the decision maker could lose confidence in a "bad" forecasting system. The results presented in this paper implicitly assumed that the decision maker's trust of the forecast was absolute. Further studies could include an explicit description of the decision maker's learning about the reliability of a forecast.



## 9 Appendix A: How the cost-loss ratio implies risk-neutrality

Consider the simple case where the decision maker has two possible choices: $s = 0$ (no action) or $s = 1$ (action). The cost of implementing the action is denoted by $c > 0$. If the adverse event occur (e.g. flood), a damage of $d > 0$ is incurred. Let also $b$ be the damage avoided if an action is taken by the decision maker (assuming $c < b \leq d$). Finally, let $p$ be the probability of the
adverse event.

Using the economic model presented in section 2.1, the vNM utility of the decision maker for each of the possible choices is:

$$U(s = 0) \quad = \quad p\mu(-d) + (1 - p)\mu(0)$$
$$U(s = 1) \quad = \quad p\mu(-d + b - c) + (1 - p)\mu(-c)$$

Straightforward algebra shows that an action is optimal (i.e. $U(s = 1) \geq U(s = 0)$) if, and only if,

$$p \geq \frac{\mu(0) - \mu(-c)}{\mu(0) - \mu(-c) + \mu(-d + b - c) - \mu(-d)} \tag{8}$$

If $\mu(\cdot)$ is concave (the decision maker is risk-averse), this is not equal to the cost-loss ratio. However, if the decision maker is risk-neutral, $\mu(\cdot)$ is linear, so for some $a_1 > 0$ and $a_2 \in \mathbb{R}$: $\mu(0) = a_2$, $\mu(-c) = -a_1 c + a_2$, $\mu(-d) = -a_1 d + a_2$ and $\mu(-d + b - c) = a_1(-d + b - c) + a_2$. Therefore, Eq. 8 reduces to:

$$p \geq \frac{c}{b}$$

If $b = d$ (all damages are avoided), this gives the usual cost-loss ratio.

Here, an important comment is in order. One could always define "cost" and "loss" as follows:

$$cost \quad = \quad \mu(0) - \mu(-c)$$
$$loss \quad = \quad \mu(0) - \mu(-c) + \mu(-d + b - c) - \mu(-d)$$

so an action is optimal if and only if:

$$p \geq \frac{cost}{loss}$$

However, this "black-box" analysis side-steps some interesting and important questions regarding the contribution of outcome versus risk preferences to the decision maker's utility. Using the vNM utility allows us to explicitly describe the impact of risk preferences on the value of forecasting systems. Note also that the hydrological literature (e.g. Roulin, 2007; Verkade and
Werner, 2011; Muluye, 2011) almost always considers "cost" and "loss" to be defined in monetary units.

## 10 Appendix B: Properties of the CARA utility function

We have: $\mu(x) = \frac{-1}{A} \exp\{-Ax\}$ for some real values for $x$ and $A \neq 0$. One can easily verify that the first derivative w.r.t. $x$ is: $\mu'(x) = \exp\{-Ax\} > 0$, and that the second derivative w.r.t. $x$ is $-A\exp\{-Ax\}$. Therefore, $\mu$ is strictly concave if $A > 0$ and strictly convex if $A < 0$.



Note that the CARA utility functions are only defined for $A \neq 0$. However, since an individual is risk-neutral if and only if $\mu$ is linear, the utility function of any risk-neutral individual has the form $\mu(x) = a_1 x + a_2$ for $a_1 > 0$ and $a_2 \in \mathbb{R}$. In other words, there is no need to define a specific class of utility for risk-neutral individuals. As such, the CARA utility class needs only to apply to non-risk-neutral individuals.

## 11  Appendix C: Simulation procedure

The simulation procedure is as follows:

1. Fix $A$ and $\psi$

2. Given the spending decision of 2014, infer the value of $\beta_{2014}$ (given the decision model).

3. Given $A$, $\psi$, $\beta_{2014}$ and the other model parameters, apply the decision-making procedure described in section 4.3 for each forecast.

4. Compute the utility, Hit Rate and Overspending indices for each forecast.

*Author contributions.*  Simon Matte performed all the computation and prepared most figures. He also wrote a preliminary version of some portions of the manuscript. Marie-Amélie Boucher initiated the project and coordinated the work. She did most of the literature review, most of the writing and prepared Figures 1, 2 and 10. Vincent Boucher proposed the economic model, prepared the Appendixes and wrote important portions of the manuscript. Thomas-Charles Fortier-Fillion provided the model and hydro-meteorological data. He participated in the interpretation of results all along the project and reviewed the manuscript.

*Competing interests.*  The authors declare that they have no conflict of interest.

*Acknowledgements.*  This work was funded by a NSERC Discovery grant to Marie-Amélie Boucher. Vincent Boucher gratefully acknowledges financial support from the *Fonds de recherche du Québec – Société et culture*. The authors wish to acknowledge Quebec's Direction of Hydrological Expertise for providing hydro-meteorological data and the model used in this study. The authors also thank the ECMWF for the development and maintenance of the TIGGE data portal allowing free access to meteorological ensemble forecasts for research purposes. Finally, this work would not have been possible without the much appreciated collaboration of Mr. Claude Pigeon, responsible for public security for the City of Quebec who, among other things, provided the economic database for the flood of 2014.





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





**Table 1.** Streamflow associated with important return periods for the Montmorency River watershed.

| Return period (years) | Streamflow ($m^3/s$) |
|:---:|:---:|
| 2 | 439.0 |
| 5 | 569.3 |
| 10 | 655.6 |
| 25 | 764.7 |
| 50 | 845.6 |
| 100 | 925.7 |
| 1000 | 1191.2 |
| 10 000 | 1456.0 |





**Table 2.** Operational flood thresholds involved in the current decision-making process for the Montmorency River watershed

| Streamflow (m$^3$/s) | Threshold | Operations |
|---|---|---|
| 350 | Surveillance | Close surveillance of river behaviour |
| 450 | Pre-Alert | Warning calls to emergency employees |
| 500 | Alert | Mobilization |
| 550 | Flood | Active evacuation |





**Table 3.** Hydro-meteorological information regarding the Montmorency River watershed.

|  | Winter | Summer |
| --- | --- | --- |
| Daily temperature (minimum/maximum °C) | -9.1 / -0.8 | 8.2 / 19.3 |
| Snowfall precipitation | 56 cm/month |  |
| Rain | 36 mm/month | 111 mm/month |





**Table 4.** Maximum fraction of total spending $s$ allowed depending of the forecasting horizon. Each spending vector is identified by an identification number (ID) for further reference.

| ID | Maximum fraction of spending allowed | | | | |
|---|---|---|---|---|---|
| Number | Day 5 | Day 4 | Day 3 | Day 2 | Day 1 |
| "No limit for a 1-day forecast" | | | | | |
| 1 | 1 | 1 | 1 | 1 | 1 |
| 2 | 0 | 0.25 | 0.5 | 0.75 | 1 |
| 3 | 0 | 0 | 0 | 0 | 1 |
| "No limit for a 5-day forecast" | | | | | |
| 4 | 1 | 0.75 | 0.5 | 0.25 | 0 |
| 5 | 1 | 0 | 0 | 0 | 0 |





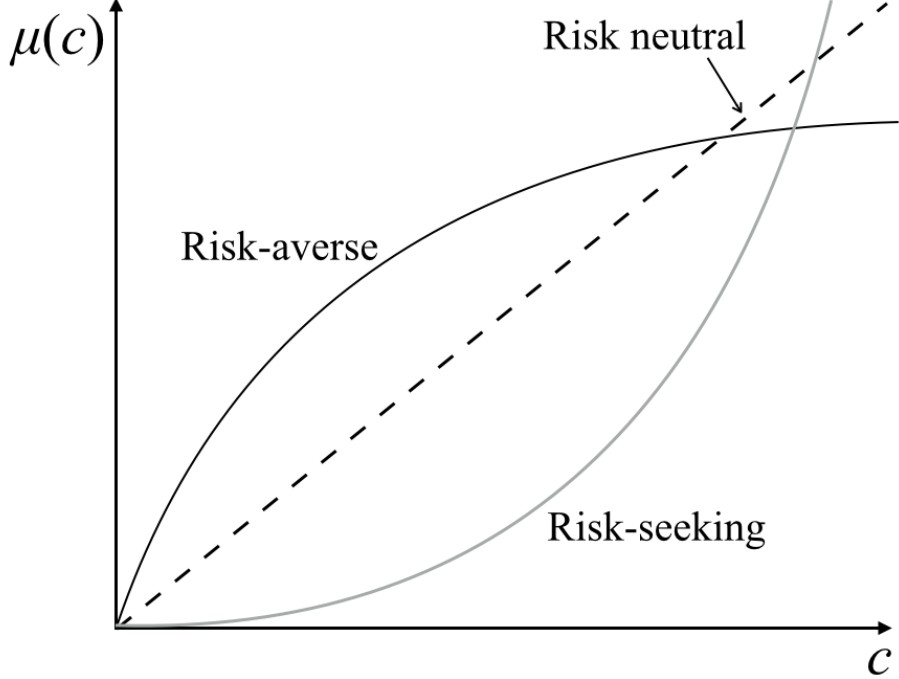

**Figure 1.** A schematic representation of the utility function for risk-averse, risk-seeking and risk-neutral individuals. For example, for a risk-averse decision maker, the concavity of the utility function implies (as an application of Jensen's inequality) that the utility of the expected outcome is greater than the expected utility of the outcome



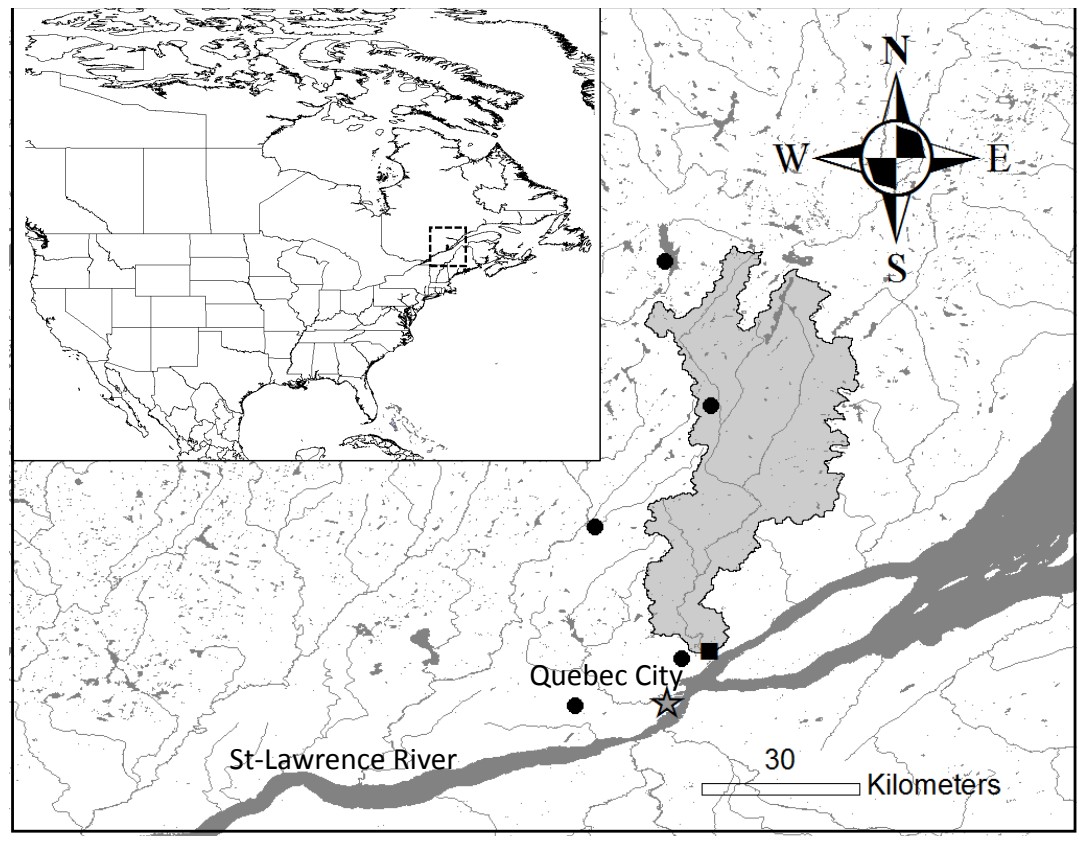

**Figure 2.** Geographical location of the Montmorency watershed. The black dots represent the available meteorological stations and the black square is the streamflow gauging station.





**Figure 3.** A portion of the 1-day and 5-day forecasted three hour time step hydrograph in 2014 against the observed streamflow; (a) and (b) are dressed forecasts, (c) and (d) are forecasts based on meteorological ensembles without EnKF and (e) and (f) are forecasts based on meteorological ensembles with state variables uncertainty estimated using the EnKF.





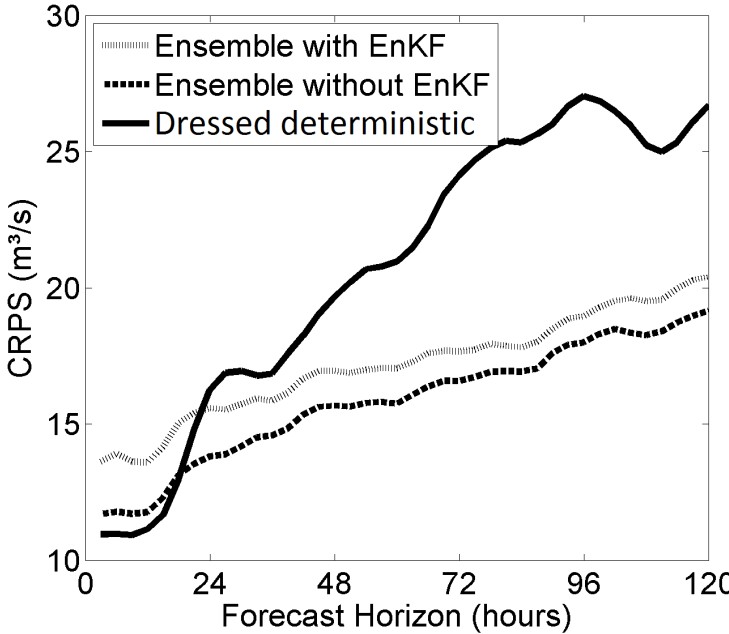

**Figure 4.** Mean CRPS as a function of lead time for the 2011-2014 period for the forecasts based on meteorological ensembles with (grey line) and without (dashed black line) state variable perturbations and for the dressed forecasts (solid black line).




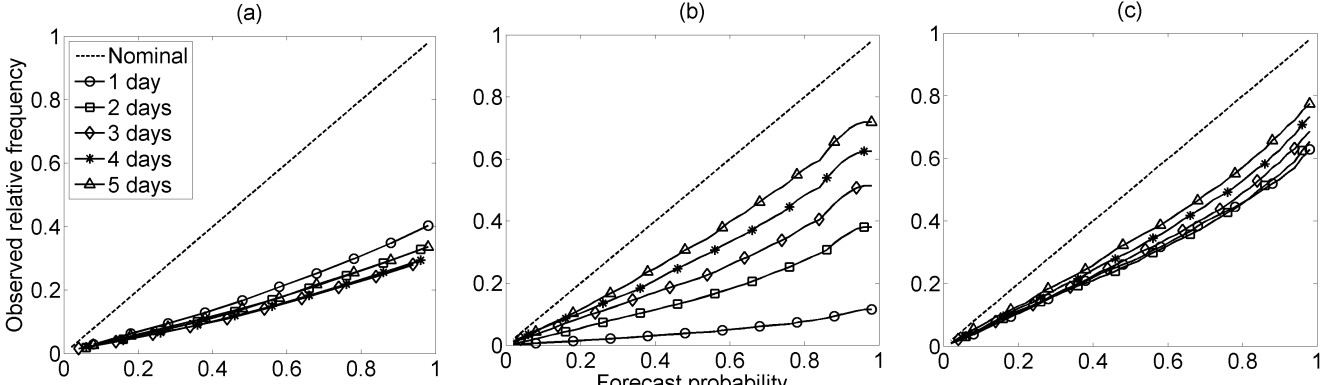

**Figure 5.** Reliability diagrams as a function of lead time for (a) dressed deterministic forecasts (b) forecasts based on meteorological ensembles and manual data assimilation and (c) forecasts based on meteorological ensembles, manual data assimilation and additional perturbation of state variables.





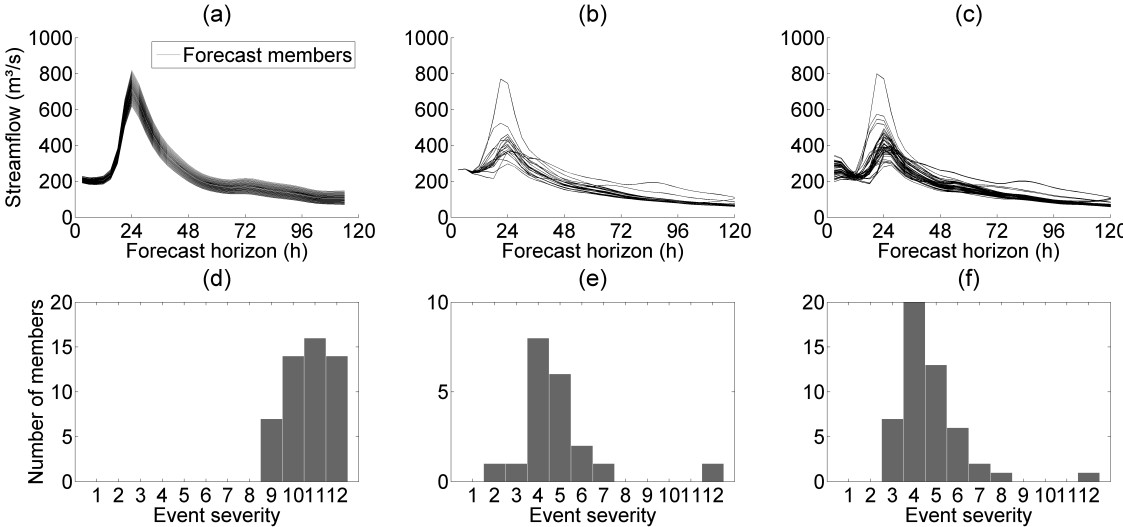

**Figure 6.** Separation of forecast members into 12 categories according to the magnitude of streamflow. The example is for forecasts emitted on May 17, 2014. (a) and (d) dressed deterministic forecasts, (b) and (e) Meteorological ensemble-based forecasts, (c) and (f) Meteorological ensemble+EnKF forecasts.





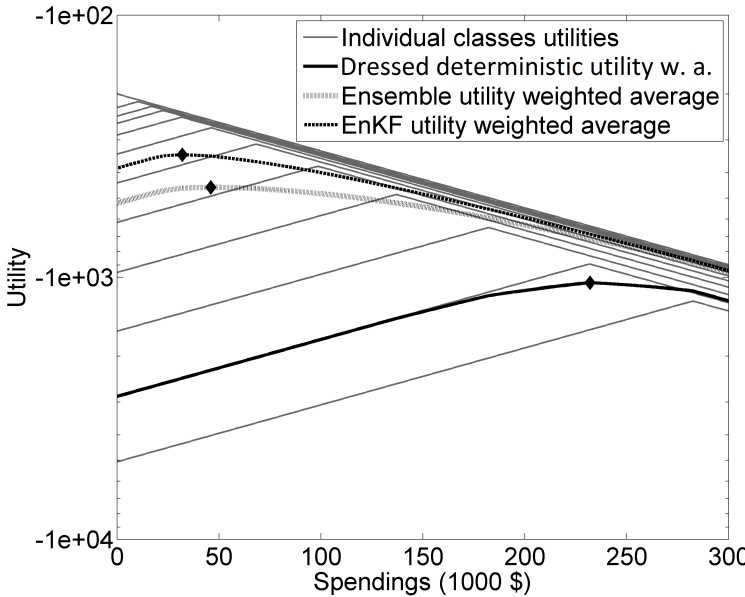

**Figure 7.** Utility as a function of money spent for forecasts emitted on May 17, 2014 for each of the three forecasting systems. Thin grey curves represent the utility of any decision given the 12 classes of events. Thick curves show the utility of forecasting system. Maxima of each system are indicated by a diamond marker. Calculations are for $A = 0.01$ and $\psi = 7$





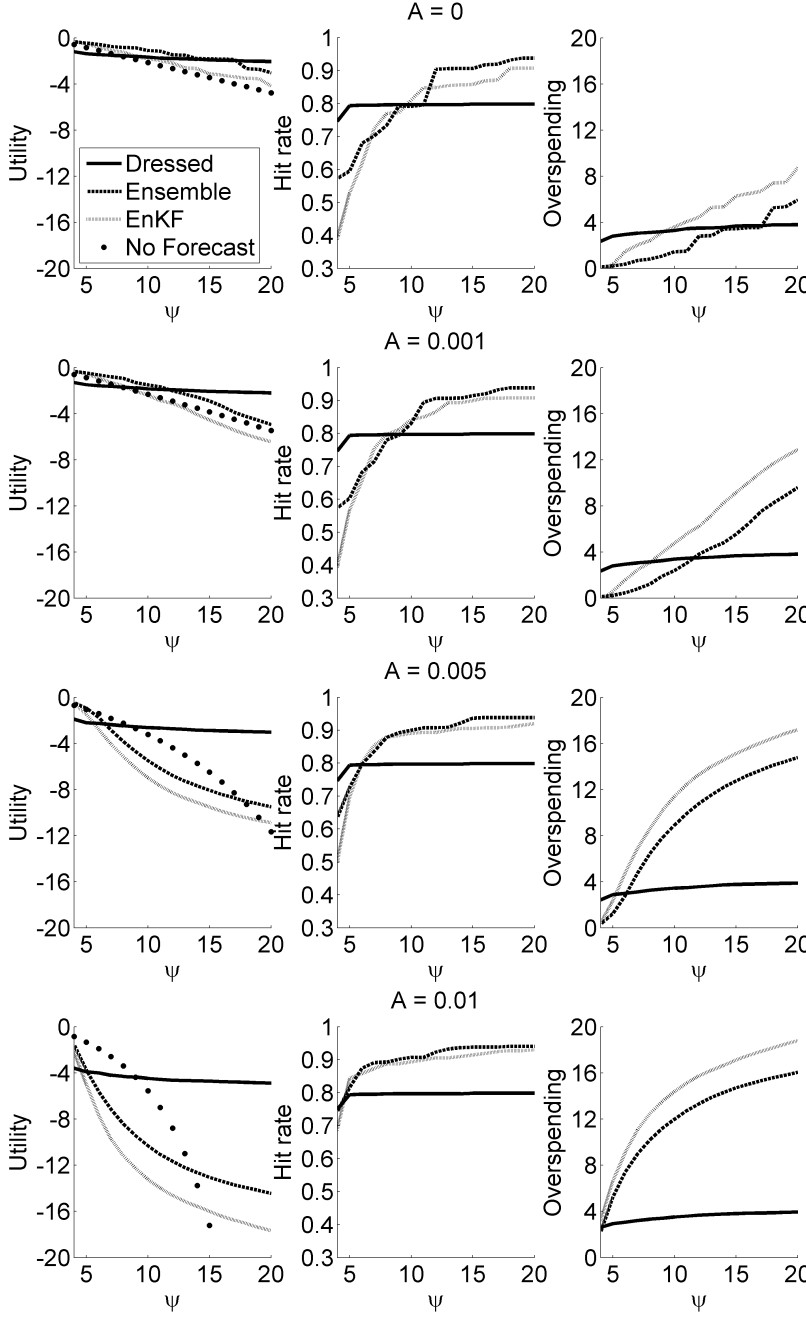

**Figure 8.** Utility, hit rate and overspending as a function of parameter $\psi$ for the three flood forecasting systems for various levels of risk aversion for the decision maker, when spending is allowed indifferently at any lead time.





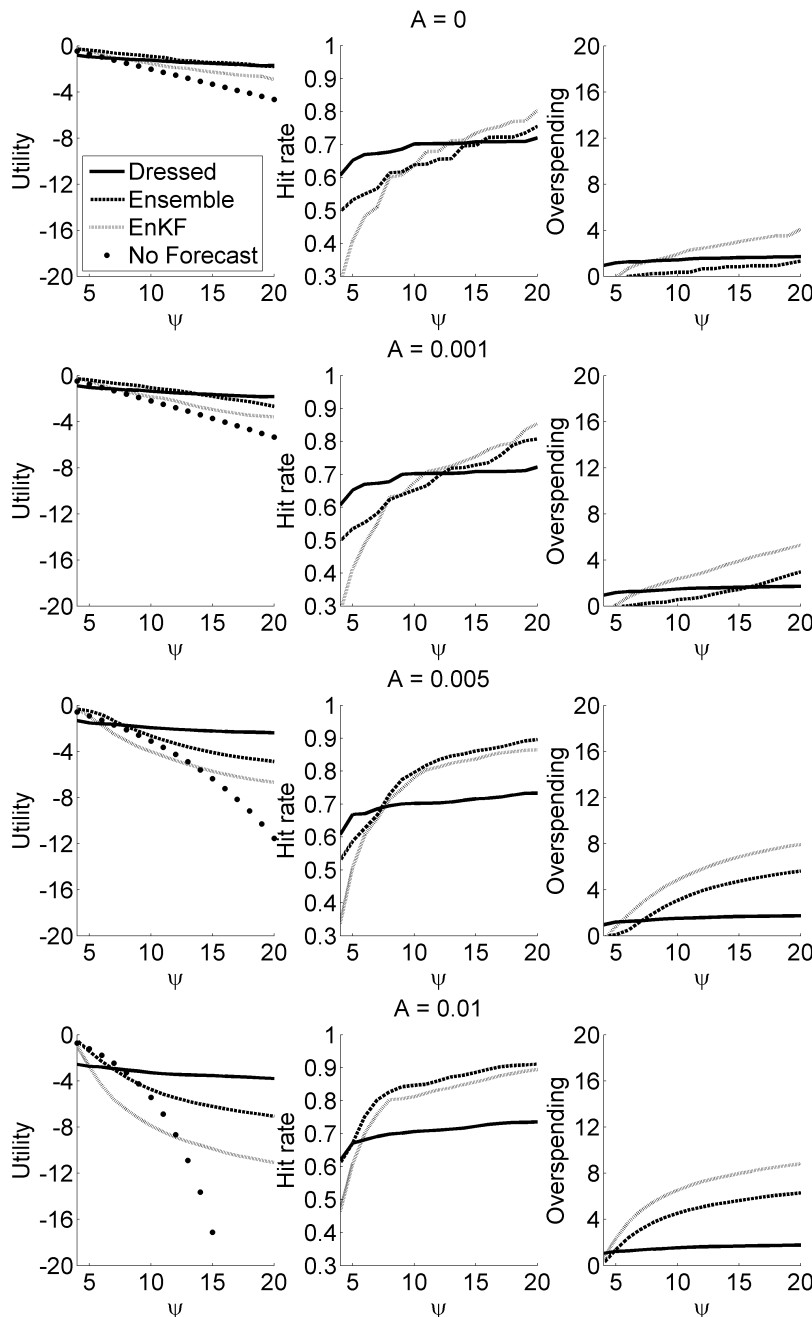

**Figure 9.** Utility, hit rate and overspending as a function of parameter $\psi$ for the three flood forecasting systems for various levels of risk aversion by the decision maker, when the decision maker is allowed to spend an increasing fraction of the total available money as the lead time shortens.





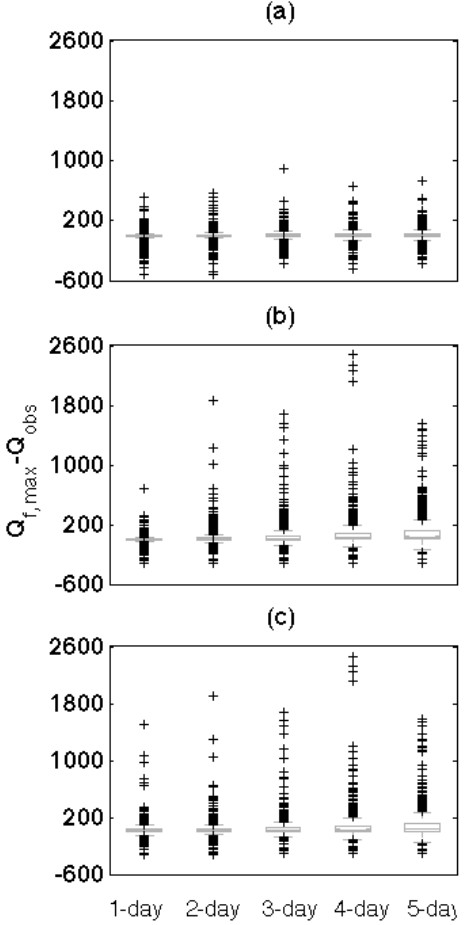

**Figure 10.** Boxplots of $Q_{f,max} - Q_{obs}$ for the three concurrent forecasting systems: (a) dressed deterministic forecasts, (b) forecasts based on meteorological ensembles, and (c) with EnKF.