# Peer review of "Moving beyond the cost-loss ratio: Economic assessment of streamflow forecasts for a risk-averse decision maker"

_Hydrology and Earth System Sciences, 2016_

## Referee Comment (RC1) · Anonymous Referee #1 · 2 Nov 2016

This is the review of the article entitled "Moving beyond the cost-loss ratio: Economic assessment of streamflow forecasts for a risk-averse decision maker" and submitted to Hydrology and Earth System Sciences by Matte et al.. It puts forward a framework for economic assessment of forecasts, in order to go beyond the cost-loss ratio analysis. It argues that this "classic" scheme does not take into account the possible risk aversion of the end-user (the decision maker). It uses the economic notion of "utility functions" which is presented as very classical in the economic field.

\*\*\* GENERAL COMMENTS \*\*\*

To the best of my knowledge, this article is quite innovative in the hydrological field. It presents a methodology intended to help answering a question which is common in

many operational forecasting centres. One important and very positive aspect of the study is that it goes beyond a simple binary decision scheme but it takes into account the fact that crisis management is a series of decisions made at different lead times. This is a clear progress in economic assessment of forecasting system.

It is rather well written and most often clear (except for a couple of sentences, see the detailed comments below). The hydrological part of the article appears to be rather sound, even if not novel (but it is not the objective of the authors).

Yet, there are 3 issues that I would like to discuss (see the specific comments section). That is why I recommend to work on sections 2 & 7.

*** SPECIFIC COMMENTS ***

A) The presentation of the economic framework is too short

Thus, this article offers an economic study to hydrologists and forecasters. It is submitted to a Journal which is mostly read by Earth scientists. All of them are not specialists of economics and therefore, an extra care should be brought to the presentation of the economic elements of the framework. As a hydrologist (not particularly aware of economic theories), I was embarrassed by the lack of explanations about the key notions and the lack of references on which I could lean on (except for the studies on the application of the cost-loss approach for flood forecasting systems which are well referenced). In my opinion, the presentation of the notion of risk aversion is too short. Risk aversion is defined as "the fact that, given the opportunity, a decision maker would be willing to spend money (or resources) to reduce the amount of uncertainty they face." (page 2, line 28). I am not convinced that the risk aversion which is discussed further (section 2...) matches this general theoretical definition. Indeed, the fact that the mu functions reflect "the decision maker's preference regarding uncertainty" (page 4) is not obvious (at least for me). As a non specialist, my naive interpretation is that it reflects the 'marginal' interest of the end user in the gain or cost. But since the uncertainty of this gain or cost is not obviously related to the gained or lost amount (it could be

as uncertain to avoid 1 M$ damage as to avoid 1k$), I don't make a clear logical link here. Furthermore, section 2 would benefit from a reorganization. Thus, appendix A is referred before the "utility" function is introduced (subsection 2.1) but it deals with this notion.

B) How is the upper tail of the predictive distribution taken into account?

There is another issue in the methodology that I don't understand completely. In section 2, the utility function is the weighted sum of the outcomes of the different "states of the world" (approximated by a (I assume sufficiently large) number of realizations of the random outcome c); these outcomes being possibly modified by the mu function. In section 4 (eq. 4), the utility is computed as the sum of the weighted outcomes of the members of an ensemble. But an ensemble, especially with a low number of members, is an approximation of the whole distribution. It does not give all the different "states of the world". How is taken into account the fact that in this study, an ensemble forecast can completely miss an event (e.g. all members can be (far) under the future observation, even if the ensemble correctly approximates the $[p=1/(N+1); p=N/(N+1)]$ part of the distribution, N being the number of members)?

In the case study, the evaluation of the benefits of the forecasts is computed over 4 years (2011-2014). I assume (I hope) that the Montmorency river basin inhabitants experienced a small number of flood events which called for mitigation actions (the authors indicate 2 floods in section 3: March 2012 and April 2014). I wonder if there is at least one flood (among all of the floods experienced during this period) for which the forecasts totally missed the event (that is, all ensemble members were (far) below the observation). If not, can we think that the results of the study would have been quite different if there were at least one missed event? Then is the data sufficient to draw full conclusions?

C) The discussion consider non scientific issues which some hydrologists and forecasters can disagree with

I have some reluctances to share completely the discussion which made me feel uncomfortable. According to me, the interpretation of the results is sometimes not crystal clear but it suggests some dangerous conclusions or behaviour in practice. The interpretation of the case study results goes far and there is a need for a stronger frame (what is the job of a forecaster? What is the job of the end user?). One main idea is that the economic value of the forecasts is the one given by the users. I agree with this idea, but aren't there two issues in practice here: - their economic value for the actual end users; - their potential economic value (for well trained users)? Both issues interest the operational forecasting community. But the answers to these questions are different and lead to different actions. I am not sure that practical conclusions should go further than:

1) risk-averse end-users mainly consider the upper tail of the predictive distribution (assessed through the largest member of the ensemble). Therefore, any 'outlier' would lead to costly actions and then the forecasts would be of low or null economic value if these outliers too often drive to false alarms. 1a) A consequence is that forecasters may be extra careful about the forecast values for high probability of non exceedance: this is a warning on the fact that equiprobable members of an ensemble (even if presented as so) can be seen with different weights by many actual end-users. In my opinion, this first conclusion should not go further than this simple warning. Otherwise, this could lead to suggestions to not present the whole predictive distribution (not all the ensemble members). Therefore, I suggest the authors to emphasize the idea in the paragraph lines 23 to 26 on page 16. It is very important not to suggest any 'biased' communication of the forecasts by the forecaster. The responsibility issue should also be discussed. Furthermore, it may be stressed that the interpretation of the ensemble can also be biased: an outlier (which is too often taken as a clue for a false alarm) is easily spotted, while the absence of any outlier (which can lead to a missed event) can not be detected! 1b) Furthermore this situation only conveys the fact that false alarms are seen as much less costly than missed events. Then, even for risk-averse users, should not the weighting of the missed events and false alarms, accordingly to their

true and known (self revealed) risk aversion be an optimal strategy? This can be done in the cost-loss ratio approach (see Verkade and Werner).

2) forecasters have to encourage and help end-users training themselves to work with sophisticated forecasts (which is a conclusion shared in the literature).

Finally, there might a confusion between uncertainty aversion and risk (of damages) aversion. A risk-averse end-user who does not know his/her degree of risk-aversion is not an optimal user (for herself or himself)? If she or he knows her/his risk aversion, why doesn't she or he quantify it (e.g. in giving comprehensive costs for false alarms and missed events)? (it is not the forecaster's job to infer them, even if she or he may help to do so) Therefore, even if this framework is interesting as an intellectual tool, it needs yet to demonstrate that it can bring more practical information than the classic cost-loss ratio methodology here.

*** DETAILED COMMENTS ***

- Page 2

Line 4: 'uncertainty *assessment of* hydrological forecasts conveys important information for decision makers' rather 'uncertainty in hydrological forecasts conveys important information ...'

Line 7: I agree that the analog forecasts are of common use. However, why quoting this approach first? Is it used by the DEH? How is it relevant for this article? I suggest the authors list the most importance uncertainty sources (in a sorted way) and then present the methodologies which can be used to deal with them. The link between analog forecasts and then ensemble forecasts (line 13) is not clear.

Line 13 ("ensemble forecasts are superior to deterministic ones"): do the authors focus on ensemble forecasts or is it true as well for probabilistic forecasts? (ensemble forecasts being used as "proto" or substitute of probabilistic forecasts, since probabilistic information is drawn from this kind of forecasts).

Lines 16-18: I agree that economic value assessment is not straightforward. However, assessing a forecast system by comparing forecasts with corresponding observations is not straightforward either. Indeed, there is not one quality but different qualities (especially for probabilistic (and then ensemble) forecasts). Different end-users would give different weights to these qualities (since they have specific applications).

Line 26 ("which does not fully exploit the information about forecast uncertainty"): what does "fully" mean here? Verkade and Werner (2011) do take explicitly into account the uncertainty.

Line 31: check spelling (Neumann / Newman)

Line 32: since the proposed framework is based on the von Neumann and Morgenstern utility function, more references are needed than a single one of 1944. Another reference is given further (page 3, line 30). But it is a book, which may be a "classic" in the economic community, but not the easiest reference to find and read by a hydrologist.

- Page 3

Lines 5 and 6: this sentence provides some conclusions of the article. Why here in the introduction?

Line 12 ("Results are presented and discussed in section 6"): in sections 6 and 7.

Line 18 ("Most importantly"): do the authors mean "More importantly"?

Line 23: check English (spelling for "weighting")

Line 30: see comment for page 2, line 32.

- Page 4

Line 5 ("the curvature of the function mu reflects the decision maker's preference regarding uncertainty"): why? Some references would be gratefully welcome.

Line 9: isn't a reference to Fig. 1 missing here?

Line 20: check English ("teh")

Line 21: check the numerotation of tables (table 3 is referred to before table 1 and 2)

- Page 5

Lines 32-33 (and lines 1-2 page 6): I did not understand why the HYDROTEL file system is useful for the reader. Are these technical details significant for this study or may they be avoided?

- Page 6

Lines 31-33: I am not sure that I understood correctly. Is the meteorological forecast ensemble used here computed by the meteorological service of Canada but taken from the TIGGE dataset (for some practical reasons)? If so, it might be clearer if stated this way.

- Page 7

Lines 10 & 11 ("Thiboult et al. (2016) showed that the [...]"): please be more specific (for this catchment? For this area?...)

Line 16: the additive coefficients for temperature inputs and the multiplicative coefficients for precipitation inputs are huge and I assume that they are much larger than the uncertainty for these inputs. Is the whole range used in practice? Is this manual 'tuning' used for more than reducing the input uncertainty in getting a best guess? Some discussion would be useful here.

Lines 19-...: the ensemble Kalman filter (as other Kalman filters) is essentially a sequential data assimilation scheme. Here, the 'update' of the M matrix is not described and it is not clear whether it is done (since this data assimilation is made after that a best guess is provided by the human forecaster). If it is not, I am not sure that this scheme may be called an ensemble Kalman filter.

- Page 8

Line 16: does 's' include the cost of the forecasting system (independently from the money spent for risk mitigation)?

Line 21: may the author provide some figures (orders of magnitude?) or some plots?

Line 28 ("these represent relatively small levels of risks of aversion"): may the authors provide some references?

Line 28 ("it is shown that they lead to qualitative changes in the decision makers"): here again, some references would help the non specialist reader.

- Page 9

Line 25: as a non specialist, I was amazed by the range of the psi factor (1.5 to 10). Is this usual?

- Page 10

Line 16: the accuracy of forecasts is inversely related to lead time. Is it inversely proportional to it?

Line 29: I am not sure that I understood the division of parameter beta_m. Why all factors (2, 1.75, 1.5,...) are larger than 1? I would have expected weights whose sum is 1.

- Page 11

Line 17: why 'then'? First results provided are the hydrographs (Fig. 3) on which doing the visual inspection.

- Page 12

Lines 24-25: I suggest that the information of appendix C comes in the main text (it is necessary for the reader).

- Page 13:

Lines 7 & 8 ("This figure shows that for 1-day forecasts, those based on meteorological ensembles and dressed deterministic forecasts have similar spread"): this is not obvious for me.

Line 16 ("For very short lead times, the dressed deterministic forecasts outperform those based on meteorological ensembles"): some discussion (interpretation) would be appreciated on this (common) behaviour.

Line 20: in practice, how does the DEH deal with the very "jumpy" ensemble curves? Are they used by operational forecasters?

- Page 14

Lines 16 & 17 ("for higher level of risk aversion [...], the decision maker SHOULD prefer the 'no forecast' situation for low levels of Psi"): doesn't the modal verb convey a notion of duty? (you are right if you do what you should do). I would rather write that the forecasting system has no (economic value) or usefulness for highly risk-averse users.

- Page 16

Line 12 ("The economic value of a forecasting system is necessarily dependent on the level of risk aversion of the decision maker"): first, it is more the economic value of the forecasts (you have to deduced its cost to get the value of the forecasting system). Then, even if I agree on the fact that it is very common (if not always), is this "necessary"? Can it be shown?

Lines 23-26: this paragraph has to be emphasized. Morevore, communicating the forecasts in a way that the end-users would perfectly understand is a key, but it is totally different from 'overforecast'.

- Page 18

Appendix A: it is referred before section 2.1 but it uses the concepts presented in this

section.

Appendix B. Where is Fig. 1 called?

- Pages 23 & 24

Tables 1 and 2 might be merged since their comparison is highly teachingful.

- Page 25

Table 3 could usefully be replaced by a plot of monthly values (if data is available)

- Page 27

Fig. 1: why is not the utility function plotted for negative values? Because if c < 0, then there is no 'interest' then the utility is 0? If so, why to use it with negative values in appendix A (for example, mu(-d))?

---

## Author Comment (AC1) · 22 Dec 2016

We are thankful for your very relevant comments, which will certainly enhance the quality of the paper. We discuss each of your specific comments below and provide our answers. In a similar fashion, we also address your detailed comments. We began preparing a revised version of the manuscript which will address theses comments. We will modify it further once we receive the comments from Reviewer 2.

[Figure]

**A) The presentation of the economic framework is too short.**

**Additional references for the presentation of economic elements, specifically risk aversion and utility theory:**

The utility theory of von Neumann and Morgenstern was proposed for the very first time in a book, not in a journal article. This might appear unconventional for hydrologists and we recognize that this might not be very convenient for the reader who wants to read this reference. Still, the generally accepted reference for utility theory is indeed von Neumann and Morgenstern (1944). We acknowledge this issue and now present additional references.

Fishburn (1989) provides a retrospective on von Neumann and Morgenstern theory with extensive excerpts from the book. Furthermore, Fishburn (1989) (which is easy to find) precisely explain the remarkable impact this theory had on the subsequent development of economic theories and also clarifies some of its limits. We will add this reference in the revised version of the manuscript. Fishburn (1989) also cites many other journal papers that the interested reader can rely on to dig deeper into utility theory. As mentioned in our manuscript, utility theory is nowadays considered "standard" in economics and its presentation can be found in numerous textbooks (e.g. Chapters 1 and 2 from Gollier (2004)) and also online. For instance, we recommend Levin (2006) and Werner (2008) which are excellent and review the main concepts (although not peer-reviewed). We will include those references in the revised version of the manuscript.

In addition to the aforementioned references for basic description of utility theory, there exists a immense amount of literature regarding applications in many different fields. For instance, Pope and Just (1991) compare different types of utility functions to represent preferences of farmers for potato acreage and also explain elements of utility theory. Although we could not find references in hydrology where risk-aversion is treated in a similar fashion (hence our contribution), we were able to find examples where the im-
portance of risk-aversion is acknowledged and described, in particular, Krzysztofowicz (1986) and Merz et al. (2009). Shorr (1966) also provides a very intuitive explanation of how the cost-loss ratio implies risk neutrality and attempts a reconciliation with utility theory in the simple context of "protect crops" vs "not protect crops". Cerdá Tena and Quiroga Gómez (2008) also explain that most decision makers are risk-averse when the stakes are high. In their paper, they illustrate how disregarding risk aversion can sometimes lead to misleading conclusions regarding the value of information (such as meteorological or hydrological forecasts). Their framework also involves the CARA utility function. However, the context of their application and the rest of their economic model is different from ours.

Since we wanted to keep the manuscript relatively short, we made the mistake of not providing enough details on the foundations of the economic framework. We agree with you that, since HESS is indeed a journal read mostly by Earth Scientists, some important details, intuitive presentations, as well as references are missing. Accordingly, we will (1) provide more detailed explanations regarding economic elements and (2) provide more references (namely Fishburn (1989), Levin (2006), Werner (2008), Krzysztofowicz (1986), Merz et al. (2009), Shorr (1966) and Cerdá Tena and Quiroga Gómez (2008). We believe that those modifications will clarify the link between risk-aversion and vNM utility functions, without introducing too much technical details and definitions, and while keeping the manuscript reasonably short. However, any additional suggestions are obviously welcome.

**About the definition of risk aversion** One of the best intuitive illustration of risk aversion and its impact on the computation of "value" is insurance. Insurance companies make sales precisely because people are risk averse. Indeed, the probability of one's house to burn down is (in general) low. You might never see your house burn down in your life (hopefully). Yet, many people buy insurance for their property. They will pay this amount of money whether or not their house really burns down (or car gets stolen or such). A risk averse person is willing to pay to increase her certainty about the future. In the context of insurance, it means that she is willing to pay a certain amount to increase her certainty of not loosing everything if a fire destroys her house, even thought he probability of this outcome is fairly low. The more risk-averse the person is, the more she is willing to pay to "remove" uncertainty or be insured.

**About the CARA utility function and reorganization of section 2**

We understand that Figure 1 is confusing. As it is in the current manuscript, Figure 1 is a generic representation of typical shapes for utility functions for risk averse, risk seeking and risk neutral individuals. It does not represent the CARA utility function. However, your comments made us realize that it would be better displaying the CARA function instead of generic shapes of utility functions. Consequently, we will reorganize section 2 according to your comments, and we will modify Figure 1 so that it is based (schematically) on the CARA utility function instead of on a generic function (see Figure 1 below).

**How $\mu$ reflects the decision maker's preferences regarding uncertainty**

In the revised version of the manuscript, we will expose with more details, both in the text and graphically (see Figure 1 below), the link between (1) the fact that risk-averse individuals would be willing to spend money in order to remove risk, and (2) the concavity of the utility function.

Consider the random variable $\tilde{c}$, and its expected value $\bar{c}$. Of course, $\tilde{c}$ has a non-degenerated probability distribution, and thus may take many possible values, while $\bar{c}$ is a known value (not risky). Now suppose that $\tilde{c}$ and $\bar{c}$ can be expressed in terms of monetary units. In the context of a lottery where players have to chose between (A) receiving a random draw $\tilde{c}$ or (B) receiving $\bar{c}$ with certainty, a risk-averse decision maker will prefer receiving $\bar{c}$ with certainty than receiving a random draw from $\tilde{c}$. That is: $U(\bar{c}) > U(\tilde{c})$, or (using the definition of vNM utility functions, as displayed on page 4 of the manuscript) equivalently $\mu(\bar{c}) > \sum_{m=1}^{M} p_m \mu(c_m)$, where $p_m$ is the probability of $c_m$. This is the mathematical definition of concavity (see also our new Figure 1 below

as well as the very intuitive explanations related to Figure 1 of Shorr (1966)).

Note that we can also define $C > 0$, the maximal amount of money that the decision maker would be willing to spend to remove the risk associated with $\tilde{c}$, as follows:

$$\mu(\bar{c} - C) = \sum_{m=1}^{M} p_m \mu(c_m).$$

That is, the individual is indifferent between receiving $\bar{c} - C$ with certainty, or receiving a random draw from $\tilde{c}$. This argument extends directly to any change in risk: any risk-averse decision maker prefers less risky distributions, in the sense of mean-preserving second order stochastic dominance (Rothschild and Stiglitz, 1970). Our new Figure 1 below also presents a graphical version of the above discussion when there are only two states of nature.

[SEE FIGURE 1 ATTACHED]
Also available online at: http://vincentbouchereconomist.com/riskaversion.pdf

**Fig. 1.** A schematic representation of the utility function for risk-averse individuals. Here, only two states of the world are assumed. The state $c_1$ is realized with probability $\alpha$ and $c_2$ is realized with complementary probability. Since $\mu$ is concave, we see that the expected utility $U(\tilde{c}) = \alpha\mu(c_1) + (1 - \alpha)\mu(c_2)$ is smaller than the utility of the expected value $U(\bar{c}) = \mu(\alpha c_1 + (1 - \alpha)c_2)$. In other words, the individual would prefer receiving the certain amount $\bar{c} = \alpha c_1 + (1 - \alpha)c_2$ than receiving a lottery $\tilde{c}$ which pays $c_1$ with probability $\alpha$ and $c_2$ with probability $1 - \alpha$. Equivalently, the individual would be willing to pay *up to* $C = \mu(\alpha c_1 + (1 - \alpha)c_2) - [\alpha\mu(c_1) + (1 - \alpha)\mu(c_2)] > 0$ to remove the risk associated with this lottery.

We will also add details in Appendix B, which describe the properties of the CARA utility function. In particular, we will show why an increase in $A$ is equivalent to an increase in the level of risk-aversion, and why the level of risk-aversion is independent of the wealth. Cerdá Tena and Quiroga Gómez (2008) also use the CARA utility function in a very simple "protect/not protect" context involving meteorological forecasts.

Finally, your are indeed right in stating that the concavity of $\mu$ "reflects the 'marginal' interest of the end user in the gain or cost". This is also true when no risk in involved as described in Shorr (1966). However, when the decision maker faces risky situations, the concavity of $\mu$ also reflects preferences toward risk (i.e. risk aversion), for reasons mentioned above.

**B) How is the upper tail of the predictive distribution taken into account?**

**"States of the world"** We acknowledge that we did not define the term "state of the world" precisely and that this might have introduce confusion. We will define more precisely what we mean by "states of the world" in the manuscript.

The set of states of the world represent the set of realizations of the random variable for which the decision maker has preferences. For instance, in Cerdá Tena and Quiroga Gómez (2008), there are only two possible states of the world: "adverse weather" and "non adverse weather". In the case of flood forecasting systems, even if streamflow values are continuous, the decision maker may only distinguish between a finite set of implied damages. In practice, this is what we use. The revised version of the manuscript will also include a figure representing the fraction of observations which fall within each damage categories. (See also below for more details.)

**Missed events in the database and sufficiency of data to draw full conclusions:**

Unfortunately, Montmorency river basin inhabitants experience a fairly high number of flood events. Many properties are located very close to the river. In our paper, we didn't want to discuss issues such as "freedom space for rivers", but clearly the question could be raised eventually.

In the paper, we assume that the decision maker has preferences over a finite number of streamflow categories, which correspond to categories of material damage, as

defined previously by Leclerc et al. (2001). We assume that the forecasted probability of occurrence of each of these categories (i.e. states of the world) is given by the fraction of the (equiprobable) members that predict streamflow values within each categories. Since the predicted streamflow values for all models cover the set of states of the world, we believe that this approximation is adequate. The revised version of the manuscript will include a new figure. It will display histograms reporting the number of events observed in each category, for each forecasting system.

Regarding missed events, there are indeed missed events in the database. Two types of missed events can be distinguished: the forecasting system can miss the *timing* of the event, the *magnitude*, or both.

Furthermore, we totally agree that the evaluation of the economic value of a forecast needs to be done over a reasonably long period of time. Technically, it relates to how well the empirical distribution of streamflow values represent the true distribution of streamflow values, and is formally represented in our context by the (empirical) expectation $\mathbb{E}_m$, as presented on page 12 of the original manuscript, and as also displayed in equations (6) and (7).

In other words, if one wants to know how "good" a forecast is, one usually looks at the *historical* performance of the forecast. This is indeed an important practical issue, which is not limited to our setting. It also affects the CRPS, as well as any other metric.

**C) The discussion consider non scientific issues which some hydrologists and forecasters can disagree with.**

We generally agree with this comment, and understand that our discussion may have carried some unwanted meaning. We will carefully rewrite the discussion accordingly. We will follow the suggestion to "emphasize the idea in the paragraph lines 23 to 26 on page 16".

We also want to make the following comments and clarifications.

1. We totally agree that there is a limit to which we should target the forecast to the end-user. In the paper, this is the reason why we didn't want to fix the level of risk-aversion. Our primary message (the upper-tail should be very well predicted) holds for any level of risk-aversion. We implicitly assume that the decision maker is "well trained" and does not suffer from (for instance) cognitive bias, and that he or she has the capacity to interpret the forecast (see also the second bullet point below). We see the training of end-users as an important, but very different issue, which we do not address in the paper.

2. We do not want to convey any indication that the forecasters should "bias" their forecast. On the contrary! Here, we assume that the decision maker completely believes the forecast. In practice, the forecast is not 100% reliable. The main message of the paper is that this is not a problem for forecast members in the lower tail of the predictive distribution. It is, however, a big problem for forecast members in the upper tail of the predictive distribution. If we simply assume that all forecast members are equiprobable (which is what is told the decision maker), when in reality, some of those are likely outliers, then we bias the predictive distribution. This problem is exacerbated for forecast members in the upper tail of the predictive distribution.

Unfortunately, we are not 100% sure to understand the last part of your third general comment (upper portion of page C5).

- **Then, even for risk-averse users, should not the weighting of the missed events and false alarms, accordingly to their true and known (self revealed) risk aversion be an optimal strategy? This can be done in the cost-loss ratio approach (see Verkade and Werner)** Our perception (perhaps wrong?)

[Figure]

is that you consider that the cost-loss ratio can allow for an evaluation of the decision maker's level of risk aversion. If so, this is not the case and this has been demonstrated in the literature (see for instance Cerdá Tena and Quiroga Gómez, 2008), as well as in Appendix A). To that effect, we will add an example following the proof in Appendix A in order to clearly expose how the predictions of the cost-loss ratio differ from the predictions of a model accounting for risk aversion.

We do not pretend that vNM utility theory is perfect. In fact, there exist a vast economic literature generalizing it and proposing alternative or complementary economic theories in contexts when it fails to convincingly predict behaviors. That being said, we believe that for the application at hand, it is the most relevant framework. Verkade and Werner (2011) is a really good paper and still one of the few to address the question of economic value of hydrological forecasts. However, it is not true that the cost-loss ratio can be used to assess the level of risk aversion of a decision maker. As written in Verkade and Werner (2011) for the description of the optimal warning rule: "It is assumed that a decision to issue a warning will only be taken if the *expected value* of the warning response is less than the *expected value* of *not* issuing a warning" (first two italics are ours). By opposition, utility theory would prescribe that "a decision to issue a warning will only be taken if the *expected utility* of the warning response is less than the *expected utility* of *not* issuing a warning". Since the utility function of a risk averse decision maker is concave, the latter statement does not lead to the same decision than the original statement from Verkade and Werner (2011). "Utility" is indeed a concept that might seem unatural at first, but it is very powerful as it encompasses more than monetary value. See for example Fishburn (1989) and Cerdá Tena and Quiroga Gómez (2008).

We believe that the economic assessment of hydrological forecasts could benefit from multidisciplinary collaborations between hydrologists and economists. This would allow for incorporation of more recent economic tools in hydro-economic

studies. However we acknowledge the complex matter of clearly exposing theories from both disciplines. We are extremely grateful for your very relevant comments which will certainly help to improve our explanations of economic theories (see answer to general comment A) to hydrologists.

- **forecasters have to encourage and help end-users training themselves to work with sophisticated forecasts (which is a conclusion shared in the literature).** We completely agree with that. However, we want to restate that risk aversion is a characteristic (of an organization or an individual) that does not disappear with training. It merely describe the preferences toward risk. In our context, a risk-neutral individual would only care about the average predicted streamflow value. This is obviously not the case, given the resources invested by forecasting organizations (includint the DEH) in order to give precise assessments of the uncertainty.

To that effect (and as a side note), we believe that forecasts should be made assuming well trained users. This is reflected in practice by our focus on vNM utility functions. There exist a well developed literature in so-called "behavioral economics" focusing on individuals' cognitive biases when working with probabilities. That is, in many important economic situations, individuals take decisions that can only be explained by their misunderstanding of probabilities (e.g. pessimism or optimism). If those theories are important to explain how individuals make decisions, we strongly believe that cognitive biases (such as pessimism) *should not* be taken into account in evaluating the value of a forecast. Our focus on vNM utility functions reflect this position since they allows to represent preferences of a well trained individual (i.e. who understand the forecasts) having (well justified) risk averse preferences.

- **A risk-averse end-user who does not know his/her degree of risk-aversion is not an optimal user (for herself or himself)? If she or he knows her/his**

[Figure]

**risk aversion, why doesn't she or he quantify it (e.g. in giving comprehensive costs for false alarms and missed events)?**

The precise measurement of risk aversion in an individual or an organization is a very complex matter and the subject of ongoing studies. Typical empirical tools such as questionnaires based on the "willingness to pay" are one possibility (i.e. finding $C > 0$ in the description of Figure 1). In our study, we have no mean of measuring the level of risk aversion of the decision maker. Because of this, and since we want our message to hold for all risk averse decision makers, we then compared our results for many values of $A$.

In fact, it does not really matter that the decision maker herself or himself doesn't know her/his level of risk aversion and it does not make this person a less "optimal" user. vNM utility function are merely a tool to explain or rationalize the decision maker's decisions. In fact, in the paper we adopt the point of view that the decision maker is a (well trained) representative of "the civil security bureau" and the level of risk aversion is that of the organization.

- **even if this framework is interesting as an intellectual tool, it needs yet to demonstrate that it can bring more practical information than the classic cost-loss ratio methodology here**

We really hope to have convinced you that the proposed framework does, indeed, bring more information than the cost-loss ratio. From our point of view, risk aversion is one such information. We added a concrete (although artificial) example in Appendix A in order to clearly explain how risk aversion affects the prediction of the cost-loss ratio.

As a side note, we would like to mention that the cost-loss ratio of the civil security for the floods on the Montmorency River seems to be greater than one (they typically spend more that the anticipated losses). This precisely makes the cost-loss ratio framework inapplicable here. Yet, when asked if they *wanted* hydrological

forecasts and if such forecasts were *valuable* (qualitatively) for them, people from the civil security responded "Yes!" without any hesitation. They also mentioned many times that they didn't regret the resources, time and money spent for flood mitigation.

The fact that C/L is greater than one can be explained by risk aversion. Even if the cost is higher than the expected loss, it is possible (i.e. there are states of the world) that the realized loss be much higher than its expected value. Risk aversion implies that the civil security is willing to spend resources to avoid those "bad" states of the world. Although we have not verified, we think possible that C/L greater than one might be more common than we think for real life situations.
**Answers to detailed comments**

**- Page 2:**
**Line 4: "uncertainty \*assessment of\* hydrological forecasts conveys important information for decision makers' rather 'uncertainty in hydrological forecasts conveys important information ..."**
Yes, of course! Thank you for pointing out this mistake.

**Line 7: I agree that the analog forecasts are of common use. However, why quoting this approach first? Is it used by the DEH? How is it relevant for this article? I suggest the authors list the most importance uncertainty sources (in a sorted way) and then present the methodologies which can be used to deal with them. The link between analog forecasts and then ensemble forecasts (line 13) is not clear.**
You are perfectly right that the mention of analog forecasts is not relevant here. All occurrences will be removed from the text. We will instead follow your suggestion of

presenting the uncertainty sources.

**Line 13 ("ensemble forecasts are superior to deterministic ones"): do the authors focus on ensemble forecasts or is it true as well for probabilistic forecasts? (ensemble forecasts being used as "proto" or substitute of probabilistic forecasts, since probabilistic information is drawn from this kind of forecasts).**
Yes it is true of probabilistic forecasts as well. The text has been modified accordingly.

**Lines 16-18: I agree that economic value assessment is not straightforward. However, assessing a forecast system by comparing forecasts with corresponding observations is not straightforward either. Indeed, there is not one quality but different qualities (especially for probabilistic (and then ensemble) forecasts). Different end-users would give different weights to these qualities (since they have specific applications).**
We fully agree. The text has been modified to reflect that.

**Line 26 ("which does not fully exploit the information about forecast uncertainty"): what does "fully" mean here? Verkade and Werner (2011) do take explicitly into account the uncertainty.**
This sentence has been rephrased in the new version of the manuscript. Verkade and Werner (2011) do indeed account for uncertainty. What we originally meant by "fully" is that they do not account for risk aversion. Therefore, the hydrological forecasts are probabilistic and account for uncertainty, but the decsion maker is assumed to be risk neutral.

**Line 31: check spelling (Neumann / Newman)**
This has been corrected in the new version of the manuscript.

[Figure]

**Line 32: since the proposed framework is based on the von Neumann and Morgenstern utility function, more references are needed than a single one of 1944. Another reference is given further (page 3, line 30). But it is a book, which may be a "classic" in the economic community, but not the easiest reference to find and read by a hydrologist.**
Indeed. Please see our answer to the general comment A.

**- Page 3:**
**Lines 5 and 6: this sentence provides some conclusions of the article. Why here in the introduction?**
This sentence was removed from the introduction in the revised version of the manuscript, according to your suggestion.

**Line 12 ("Results are presented and discussed in section 6"): in sections 6 and 7. Line 18 ("Most importantly"): do the authors mean "More importantly"?**
This has been corrected in the new version of the manuscript.

**Line 23: check English (spelling for "weighting")**
This has been corrected in the new version of the manuscript.

**Line 30: see comment for page 2, line 32.**
Done.

**- Page 4:**
**Line 5 ("the curvature of the function mu reflects the decision maker's preference regarding uncertainty"): why? Some references would be gratefully**

**welcome.**
Indeed. Please see our answer to the general comment A.

**Line 9: isn't a reference to Fig. 1 missing here?**
Please also see our answer to the general comment A. Figure 1 was initially meant as a generic illustration of utility functions for risk neutral, risk seeking and risk averse individuals. It is now based on the CARA utility function.

**Line 20: check English ("teh")**
This has been corrected in the new version of the manuscript.

**Line 21: check the numerotation of tables (table 3 is referred to before table 1 and 2)**
This has been corrected in the new version of the manuscript.

**- Page 5:**
**Lines 32-33 (and lines 1-2 page 6): I did not understand why the HYDROTEL file system is useful for the reader. Are these technical details significant for this study or may they be avoided?**
The aforementioned lines have been removed from the modified version of the manuscript, according to your suggestion.

**- Page 6:**
**Lines 31-33: I am not sure that I understood correctly. Is the meteorological forecast ensemble used here computed by the meteorological service of Canada but taken from the TIGGE dataset (for some practical reasons)? If so, it might be clearer if stated this way.**

As strange as it might seem, it is indeed much easier to obtain an archive of past Canadian ensemble forecasts through TIGGE than directly from Environment and Climate Change Canada. This is due to a number of practical reasons that we prefer not to detail here. However, according to your suggestion, the text of the manuscript was modified and now reads: "Precipitation and temperature ensemble forecasts from the Meteorological Service of Canada (MSC) covering the 2011—2014 period are used. For practical reasons, those forecasts were obtained from the Thorpex Interactive Great Grand Ensemble (TIGGE) database managed by the European Center for Medium Range Weather Forecasts (ECMWF)."

**- Page 7:**
**Lines 10 & 11 ("Thiboult et al. (2016) showed that the [...]"): please be more specific (for this catchment? For this area?...)**
The work of Thiboult et al. (2016) was performed on 20 catchment in Quebec. The Montmorency River was not included in this, but it is located in the same general area (Quebec, Canada). The manuscript was modified to include this precision: "In a study involving 20 catchments in Quebec, Thiboult et al. (2016) showed that the uncertainty for initial conditions dominates ..."

**Line 16: the additive coefficients for temperature inputs and the multiplicative coefficients for precipitation inputs are huge and I assume that they are much larger than the uncertainty for these inputs. Is the whole range used in practice? Is this manual 'tuning' used for more than reducing the input uncertainty in getting a best guess? Some discussion would be useful here.**
Yes they are huge. They are the true operational limits at the DEH. However, it is worth emphasizing that the goal of those perturbations on precipitations and temperature is to (indirectly) affect state variables (soil moisture, snow water equivalent) and correct model uncertainties. They are not intended as to reflect the true uncertainty on

precipitations and temperature. The goal of this manual tuning is indeed to obtain a best guess regarding the initial state of the watershed (under the assumption that the state variables of the model accurately reflect the state of the watershed). However, it might reassure the reviewer (as well as everybody else) to know that those huge limits for perturbations are rarely reached. In our study, the multiplicative coefficient applied to precipitation varied between 0.5 and 2.5. Most additive coefficients for temperature varied between -3 and +2.5, with occasional large coefficient (up to -7 and +7 on 2-3 occasions). Precisions regarding what perturbations were really applied and the limits that were permitted were added in the revised version of the manuscript.

**Lines 19-...: the ensemble Kalman filter (as other Kalman filters) is essentially a sequential data assimilation scheme. Here, the 'update' of the M matrix is not described and it is not clear whether it is done (since this data assimilation is made after that a best guess is provided by the human forecaster). If it is not, I am not sure that this scheme may be called an ensemble Kalman filter.**
The EnKF that is implemented here follows Thiboult et al. (2016) and Mandel (2006). $M$ is the model error covariance matrix, computed before data assimilation, at each time step of the sequential data assimilation. As such, it is not updated, as it is the model's state variables that are updated according to equation (3). Of course, updating the state variables will affect the model outputs, hence M at the following time step. Thus, in our specific implementation state variables are indeed updated, but not from an open loop simulation. The base line simulation here is the manually assimilated run. This base line simulation is good, but cruelly lacks dispersion. In that context, the purpose of the EnKF is only to consider uncertainty associated to state variables and not to improve the first guess estimate of state variables. The parameters of the EnKF were not fine tuned as in many studies (such as Thiboult and Anctil, 2015), for instance. Random perturbations added to temperature were drawn from uniform distributions U[-8,+8]° and U[0.5, 1.5] (multiplicative) for precip. This choice is coherent with the way the manual data assimilation was performed, but could certainly

be improved. For instance, normal error distributions are most commonly used and the spread of those distribution is calibrated until good agreement with observation is achieved. Again, the goal of the EnKF here is to add spread around best estimate of state variables, in a controlled and systematic manner. We consider that further refinement of the EnKF is outside the scope of our study.

**- Page 8:**
**Line 16: does 's' include the cost of the forecasting system (independently from the money spent for risk mitigation)?**
No, it does not. First, we unfortunately don't have this information. Second, and perhaps more importantly, this not the objective of the paper. We focus on the economic value of different forecasts. Of course when the civil security chooses which forecasting system to put in place, it must take into account the value of the system (i.e. how it will affect future spending decisions, and resulting utilities) as well as its cost. The important point is that once the system is in place, its cost should not affect spending decisions. This also motivated our focus on CARA utility functions since they do not depend on "wealth" (which would be affected by the cost of performing the forecast).

**Line 21: may the author provide some figures (orders of magnitude?) or some plots?**
Unfortunately we probably can't, as these are confidential. We asked civil security of Quebec for the permission to provide figures in terms of percentages or such and we are waiting for their answer.

**Line 28 ("these represent relatively small levels of risks of aversion"): may the authors provide some references?**
Indeed, we added a reference (Babcock et al., 1993) which provide a review of many

assumed levels of risk-aversion in the literature.

**Line 28 ("it is shown that they lead to qualitative changes in the decision makers"): here again, some references would help the non specialist reader.**
We meant "in this paper". We will add a precision in order to avoid confusion. We see from Figures 8 and 9, that a departure from $A = 0$ strongly affects the comparison between the three forecasting systems.

**- Page 9:**
**Line 25: as a non specialist, I was amazed by the range of the psi factor (1.5 to 10). Is this usual?**
The range of $\psi$ captures two important aspects. First, in 2014, the civil security spent around 3.5 times more than the realized material damages. This reflects the fact that (perhaps obviously) the decision maker also consider immaterial damages. Since it is extremely hard to evaluate immaterial damages, we let $\psi$ vary to (very) large values. We actually also performed simulations for values much higher than 10. They are not displayed in the current version of the manuscript as the analysis would remain the same, but the graphs would be harder to read.

We believe that $\psi = 10$ is a reasonable value. Recall that immaterial damages include any damage that cannot be easily expressed in monetary value. Those include losses in the "quality of life", avoiding law suits (including the associated bad press)... and can therefore be quite high.

**- Page 10:**

**Line 16: the accuracy of forecasts is inversely related to lead time. Is it inversely proportional to it?**

No, in general they are not.

**Line 29: I am not sure that I understood the division of parameter $\beta_m$. Why all factors ($2, 1.75, 1.5, ...$) are larger than 1? I would have expected weights whose sum is 1.**
Those factors reflect the benefit of early warning. The baseline is the 1-day ahead warning, so any early warning should be more beneficial. In practice, this reflects the fact that the population has time adjust (pack, empty their basements, arrange visits to their relatives...) before being evacuated.

**- Page 11:**
**Line 17: why 'then'? First results provided are the hydrographs (Fig. 3) on which doing the visual inspection.**
This sentence has been modified. It now reads: "Firstly, a visual inspection of the forecasted hydrographs is undertaken. This performance assessment also involves the well-known Continuous Ranked Probability Score (...)"

**- Page 12:**
**Lines 24-25: I suggest that the information of Appendix C comes in the main text (it is necessary for the reader).**
The content of Appendix C has been placed at the very end of section 4.3 in the revised version of the manuscript. It is introduced by "To summarize, the simulation procedure is as follows:"

**- Page 13:**
**Lines 7 & 8 ("This figure shows that for 1-day forecasts, those based on meteorological ensembles and dressed deterministic forecasts have similar spread"):**

**this is not obvious for me.**
Yes, indeed. Without getting into too much detail, this mistake is an artifact from a previous version of the manuscript. The beggining of section 6.1 has been modified. The sentence now reads "This figure shows that for 1-day forecasts, forecasts based on meteorological ensembles generally have low spread. This is expected, as only the forcing uncertainty is accounted for and this uncertainty requires more than one day to be propagated through the hydrological model. In addition, at short lead times the members of meteorological ensemble forecasts are often very similar. However, before each of the two flood peaks, they display more dispersion than dressed forecasts."

**Line 16 ("For very short lead times, the dressed deterministic forecasts outperform those based on meteorological ensembles"): some discussion (interpretation) would be appreciated on this (common) behaviour.**
The following sentence was added to the revised version of the manuscript: "As noted above, for short lead times the members of the meteorological ensemble forecasts are often very similar and the forecasts thus have no dispersion. Dressed forecasts, by definition, necessarily have more spread. Since the forecasting system is not perfect, an ensemble with very low spread is at risk of missing the observation."

**Line 20: in practice, how does the DEH deal with the very "jumpy" ensemble curves? Are they used by operational forecasters?**
No. The DEH doesn't use forecasts based on meteorological ensembles. They use the dressed deterministic forecasts, which are not so "jumpy". This is mentioned on page 3 (line 73-74).

**- Page 14:**
**Lines 16 & 17 ("for higher level of risk aversion [...], the decision maker SHOULD prefer the 'no forecast' situation for low levels of Psi"): doesn't the modal verb**

[Figure]

**convey a notion of duty? (you are right if you do what you should do). I would rather write that the forecasting system has no (economic value) or usefulness for highly risk-averse users.**

Yes, we agree. The text has been changed to include this suggestion.

**- Page 16:**

**Line 12 ("The economic value of a forecasting system is necessarily dependent on the level of risk aversion of the decision maker"): first, it is more the economic value of the forecasts (you have to deduced its cost to get the value of the forecasting system). Then, even if I agree on the fact that it is very common (if not always), is this "necessary"? Can it be shown?**

The text has been changed (forecasts instead of forecasting system). This specific comment is in line with general comments A-B-C. We added many new references supporting the importance of considering risk aversion in the evaluation of the economic value of forecasts.

**Lines 23-26: this paragraph has to be emphasized. Morevore, communicating the forecasts in a way that the end-users would perfectly understand is a key, but it is totally different from 'overforecast'.**

We agree. As mentionned above (see answer to general comment C), this portion of the manuscript will be improved thanks to the reviewer's comments.

**- Page 18**

**Appendix A: it is referred before section 2.1 but it uses the concepts presented in this section.**

We thank the reviewer for pointing out this inconsistency. However, we think that it is important to leave the reference to Appendix A in section 2 since it is where we explain the limits of the cost-loss ratio. We also don't want the reader to think that we

make unsupported claims. Consequently, line 20 now reads "Appendix A illustrates a technical presentation that builds on the concepts presented in the next section."

**Appendix B. Where is Fig. 1 called?**
Figure 1 was modified to avoid confusion and now represents the CARA utility function. Text has been modified accordingly

**- Pages 23 & 24:**
**Tables 1 and 2 might be merged since their comparison is highly teachingful.**
We agree. This will be included in the revised version of the manuscript

**- Page 25**
**Table 3 could usefully be replaced by a plot of monthly values (if data is available)**
We agree. This will be included in the revised version of the manuscript

**- Page 27:**
**Fig. 1: why is not the utility function plotted for negative values? Because if c < 0, then there is no 'interest' then the utility is 0? If so, why to use it with negative values in appendix A (for example, mu(-d))?**
Figure 1 now includes negative values of $c$. Appendix A has also been improved. Indeed, one of the advantages of working with CARA utility functions is that they are defined for any value of $c$, and not just for positive values.

**References**

Babcock, B. A., Choi, E. K., and Feinerman, E.: Risk and probability premiums for CARA utility functions, Journal of Agricultural and Resource Economics, pp. 17–24, 1993.

Cerdá Tena, E. and Quiroga Gómez, S.: Cost-Loss Decision Models with Risk Aversion, vol. 2008, Instituto Complutense de Estudios Internacionales, 2008.

Fishburn, P.: Retrospective on the Utility Theory of von Neumann and Morgenstern, Journal of Risk and Uncertainty, 2, 127–158, 1989.

Gollier, C.: The economics of risk and time, MIT Press, 2004.

Krzysztofowicz, R.: Expected utility, benefit, and loss criteria for seasonal water supply planning, Water Resources Research, 22, 303–312, 1986.

Leclerc, M., Morse, M., Francoeur, J., Heniche, M., Boudreau, P., and Secretan, Y.: Analyse de risques d'inondations par embâcles de la rivière Montmorency et identification de solutions techniques innovatrices – Rapport de la Phase I – Préfaisabilité, Tech. Rep. R577, INRS-Eau and Laval University, Quebec, 2001.

Levin, J.: Choice under uncertainty, Lecture Notes, http://web.stanford.edu/%7Ejdlevin/Econ%20202/Uncertainty.pdf, 2006.

Mandel, J.: Efficient implementation of the Ensemble Kalman Filter, Tech. Rep. R1416, University of Colorado at Denver and Health Sciences Center, Denver, 2006.

Merz, B., Elmer, F., and Thieken, A.: Significance of "high probability/low damage" versus "low probability/high damage" flood events, Natural Hazards and Earth System Sciences, 9, 1033–1046, 2009.

Pope, R. and Just, R.: On testing the structure of risk preferences in agricultural supply analysis, Agricultural Journal of Agricultural Economics, 73, 743–748, 1991.

Rothschild, M. and Stiglitz, J. E.: Increasing risk: I. A definition, Journal of Economic theory, 2, 225–243, 1970.

Shorr, B.: The cost/loss utility ratio, Journal of Applied Meteorology, 5, 801–803, 1966.

Thiboult, A. and Anctil, F.: On the difficulty to optimally implement the Ensemble Kalman filter: An experiment based on many hydrological models and catchments, Journal of Hydrology, 529, 1147–1160, 2015.

Thiboult, A., Anctil, F., and Boucher, M.-A.: Accounting for three sources of uncertainty in ensemble hydrological forecasting, Hydrology and Earth System Science, 20, doi:10.5194/hess-20-1809-2016, 2016.

von Neumann, J. and Morgenstern, O.: Theory of games and economic behavior, vol. 60, Princeton University Press Princeton, 1944.

Werner, J.: risk aversion, in: The New Palgrave Dictionary of Economics, edited by Durlauf, S. N. and Blume, L. E., Palgrave Macmillan, Basingstoke, 2008.

---

## Referee Comment (RC2) · Anonymous Referee #2 · 31 Dec 2016

The paper introduces the application of the Constant Absolute Risk Aversion (CARA) utility function in flood warning systems. The function enables the consideration of risk-averse behaviour of decision makers. The paper thus presents new and innovative scientific work. In general, the paper is well organised and well written. In the revised version, a few things should be changed.

Abstract:

- No abbreviations should be used in the abstract without explanation.

- A sentence summarising the main conclusion of the work should be added.

Introduction:

[Figure]

- Again, abbrevations are not well explained. Please provide the full term when the abbreviation is used for the first time. Please check in the whole paper.

- p. 2, line 7/8: What does this mean? Could you provide examples?

- In general, a more structured review of the literature on uncertainty is missing. For example, different types of uncertainty (epistemic versus aleatoty/natural uncertainty) could be distinguished since they may have different effects on decisions and decision makers because epistemic uncertainty can be reduced by better data or models while aleatory uncertainty cannot. Later in the paper, this should also be discussed in the context of the study.

- The von Newman and Morgenstern utility function should already be briefly explained in the introduction (p. 2, line 31/32).

- p. 3, line 2: delete "forecast" once.

Section 2

- If you use a section 2.1 there should also be a section 2.2. One subheading does not make sence. Consider to delete the headline.

- The economic model and the utility functions should be better explained. The content of the chapter referenced in line 30 (p. 3) should be briefly summarized.

- A paragraph that bridges this section to the next should be added.

- Starting on p. 4: Check the numbering of the equations; add numbers to all equations on p. 4, 9 and 12.

Section 3

- Typo in line 20 (p. 4): THE

- p. 4, line 28/29: consider rephrasing, check logic

- In Table 1, the potential damage should be added fpr each return period.

- p. 5, line 8/9: consider rephrasing ("cause" is used twice in this short sentence)

- p. 5, line 31: This is unclear. The calibration performed by DEH should be explained (as well as the meaning of DEH - see my comment on the use of abbreviations)

- Again, there shouldn't be a section 3.3.1 only. Please re-organise the text.

Section 4

- p. 8, line 14: The use of 12 categories should be justified or better explained.

- p. 8, line 19-21: The content and use of the data for the 2014 flood is unclear. Please add some information.

- p.9, line 4-6: The basis/source of the mentioned losses is unclear. Please explain how these values were derived. In line 27, a damage curve of Leclerc et al. (2001) is mentioned. This comes too late and too vague. Explain how the curve looks like, whether it is applicable in the catchment under study or/and whether and how is was adpated to your case study.

- p. 9, line 4 and line 10: consider using "losses" instead of "damages".

- p. 10, line 3 to 15: Most of this should be shifted to the dicsussion section.

- In general, the section 4.3 is somewhat unclear and contains too many issues for dicsussion. Consider to shorten it to the main point that are necessary for the model application.

Section 5

- p. 12, line 3: discuss how the true distribution of streamflow could be determined or whether it is possible to check the validity of the used distribution.

Section 6

- p.13, line 22: What do you mean by "sharpness"? Accuracy?

- p. 14, line 23: What do you refer to when you mention "relatively rare and comparatively small flood events"?

Section 7: —

Section 8: p. 17, line 19: typo "AND in terms.."

Figure 3, 4 and 10: Explain the abbreviation in the figure caption.

---

## Author Comment (AC2) · 16 Jan 2017

We thank you very much for your comments and suggestions which will help us to improve the manuscript. Some of those comments and suggestions rejoin some concerns also expressed by Reviewer 1. In the following, we address each comment and suggestion. In some cases, we refer to our (rather lengthy) response to Reviewer 1.

**Abstract**

**- No abbreviations should be used in the abstract without explanations.**

**- A sentence summarizing the main conclusions of the work should be added.**

We will replace CARA by "Constant Absolute Risk Aversion" in the abstract, according to your suggestion. As for the addition of " A sentence summarising the main conclusion of the work", it was precisely the goal of the last sentence of the abstract: "It is found that the economic value of a forecast for a risk-averse decision maker is closely linked to the forecast reliability in predicting the upper tail of the streamflow distribution." We will also add a sentence clarifying the impact of this finding for the design of future forecasts.

**Introduction**

- **Again, abbreviations are not well explained. Please provide the full term when the abbreviation is used for the first time. Please check the whole paper.**
  We verified the entire manuscript for acronyms and obtained the following list (the pages and line numbers refer to version of the manuscript initially submitted):

  - CARA: Constant Absolute Risk Aversion. Appears for the first time in the abstract and **will be defined there in the revised version of the manuscript**.
  - HEPEX: Hydrological Ensemble Prediction EXperiment. Appears for the first time at line 1 of the introduction and **will be defined there in the revised version of the manuscript**.

- DEH: Direction de l'Expertise Hydrique. Appears for the first time at line 3 of page 3 and is already defined there.
- vNM: von Neumann and Morgenstern. Appears for the first time at line 32 of page 2 and is already defined there. Typo in the "Neumann" pointed out by Rev. 1 was corrected in the revised version of the manuscript.
- CRPS: Continuous Ranked Probability Score. Appears for the first time at line 31 of page 11 and is already defined there.
- INRS: Institut National de la Recherche Scientifique. Appears for the first time at line 17 of page 5 and is already defined there.
- RHHU: Relatively Homogenous Hydrological Unit. Appears for the first time at line 21 of page 5 and is already defined there.
- SWE: Snow Water Equivalent. Appears only once, on page 6 line 2 and is defined there. **We might remove the acronym since we are not using it later**.
- BV3C: *Bilan Vertical en 3 Couches*. Appears only once at line 28 of page5. **Will be defined**. It is the name of a subroutine of HYDROTEL
- TIGGE: THORPEX Interactive Grand Global Ensemble. THORPEX is itself an acronym, which stands for "The Observing system Research and Predictability Experiment". Appears for the first time at page 6, line 31. The acronym TIGGE is already defined there, but not THORPEX. **The definition of THORPEX will be added**.
- ECMWF: European Center for Medium Range Weather Forecasting. Appears for the first time at line 32 of page 7 and is already defined there.
- MSC: Meteorological Service of Canada. Appears for the first time at line 33 of page 6 and is already defined there.
- EnKF: Ensemble Kalman Filter. Appears for the first time at line 21 of page 7 and is already definied there.

- – USACE: United States Army Corps of Engineers. Appears only at line 30 of page 10 and in the list of reference. **It will be modified so that the full name appears instead of the acronym, both in the text and in the list of references**.

- **p. 2, line 7/8: What does this mean? Could you provide examples?** According to a comment by Reviewer 1, all references to analog forecasting systems have been removed from the revised version of the manuscript.

- **In general, a more structured review of the literature on uncertainty is missing. For example, different types of uncertainty (epistemic versus aleatoty/natural uncertainty) could be distinguished since they may have different effects on decisions and decision makers because epistemic uncertainty can be reduced by better data or models while aleatory uncertainty cannot. Later in the paper, this should also be discussed in the context of the study.**

Identifying the different types of uncertainty can certainly help guiding the improvement of forecasts. This indeed has been the focus of many papers (e.g. Juston et al., 2013; Beven, 2016). The later reference includes a section about the level of *confidence* that one can have in the forecasts and how this level of confidence (potentially affected by disinformation and uncertainty) can impact decision-making. This is briefly discussed on page 17, line 30 of the current version of the manuscript. In the present study, which is only the first step toward a more realistic assessment of the value of forecasts, we assume that "the decision maker's trust of the forecasting system is absolute" (p.17 line 31-32). Considering the level of confidence of decision-maker has toward the forecasting system, and how this level of confidence can vary according to his/her ongoing experience with the forecasts, requires further significant modifications of the decision model. As mentioned on p. 17 line 32-33, this will be the object of future work.

Now, under this assumption that "the decision maker's trust of the forecasting system is absolute", the identification and reduction of different sources of uncertainty is somewhat distinct from the decision maker's problem. For a given forecast, the decision maker's spending decision does not depend on the *type* of uncertainty. The decision maker takes the forecast as given, and does not directly take part in its elaboration. This is also linked to Reviewer 1's comment C (Page 8, line 16 of the original manuscript) regarding the "cost of the forecasting system". Any decision that may affect the quality of the forecasts (implementation of new forecasts, reduction of uncertainty...) has been taken before the decision maker has to decide how much to spend. At this point, he or she takes the forecast as such and makes the best possible decision given the information available at that time.

Given your comment, as well as Reviewer 1's, we will adjust the discussion in order to avoid such confusion. We will also include the above mentioned references regarding the types of uncertainty in the revised version of the manuscript.

- **The von Neumann and Morgenstern utility function should already be briefly explained in the introduction (p. 2, line 31/32)** The vNM utility functions will be intuitively described in the introduction of the revised version of the manuscript. We will also add many details in section 2. See our detailed answer to comment A from Reviewer 1.

- **p. 3, line 2: delete "forecast" once.**: Will do. Thank you.

**Section 2**

- **If you use a section 2.1 there should also be a section 2.2. One subheading does not make sence. Consider to delete the headline.** The headline of section 2.1 will be deleted according to your suggestion.

- **The economic model and the utility functions should be better explained. The content of the chapter referenced in line 30 (p. 3) should be briefly summarized.** Utility theory and the economic model will indeed be better explained in the revised version of the manuscript. Please see our answer to Comment A by Reviewer 1. He/she also had many questions and comments regarding those topics and asked for additional references.

- **A paragraph that bridges this section to the next should be added** A paragraph (or rather a sentence) that bridges this section to the next will be added according to your suggestion.

- **Starting on p. 4: Check the numbering of the equations; add numbers to all equations on p.4,9 and 12.** All equations will be numbered in the revised version of the manuscript.

**Section 3**

- **Typo in line 20 (p. 4)** Will be corrected, thank you.

- **p. 4, line 28/29: consider rephrasing, check logic** Those lines currently read: "The response time of the watershed is rapid (12 hours). The return period of damaging floods is also short. This makes emergency evacuation and flood damage a common occurrence for riverside". We would like more precisions on what to clarify. First sentence means that floods appear rapidly. Second sentence means that floods happen often. This is why it is important to have flood forecasts and an emergency plan for this particular watershed.

- **In Table 1, the potential damage should be added fpr each return period.** According to our answer to Reviewer 1's comment B (second item), the revised

version of the manuscript will include histograms reporting the number of events observed in each category of damages, for each forecasting system. We believe that this information will be more in line with the general framework of the paper. In addition, we are worried that displaying the values derived from the flow-damage curve provided in Leclerc et al. (2000), which would be gross approximations, could lead the reader to put too much confidence in those estimates.

Note also that Leclerc et al. (2000)'s report (though in French) is freely accessible on the Internet.

- **p. 5, line 8/9 consider rephrasing ("cause" is used twice in this short sentence)** The sentence will be rephrased to read: "an early spring thaw caused by extreme temperatures **induced** a flood resulting in the evacuation of 25 households."

- **This is unclear. The calibration performed by the DEH should be explained (as well as the meaning of DEH - see my comment on the use of abbreviations)** The meaning of "DEH" is already defined at the first use of this acronym (please see our answer to your comments about the introduction). The DEH is a section of our provincial government (province of Quebec) that is responsible for hydrology and hydraulics (all aspects: operational flood forecasting, data collection and dissemination, dam safety control, etc.).

The calibration of model's parameters was performed using the Shuffle Complex Evolution algorithm of the University of Arizona (SCE-UA, Duan et al., 1994). The objective function to maximize is the Nash-Sutcliffe Efficiency criterion. We propose to mention briefly SCE-UA, Nash-Sutcliffe efficiency criterion and global calibration strategy in the revised version of the manuscript. We assume SCE-UA and the Nash-Sutcliffe efficiency to be well understood by most readers.

- **Again, there shouldn't be a section 3.3.1 only. Please reorganize the text.**

We agree and section 3.3.1 will be removed. The content will simply be merged with 3.3.

**Section 4**

- **p. 8, line 14: The use of 12 categories should be justified or better explained.** First, we wanted to separate streamflow values into different categories because the result of a 450 m3/s flood shouldn't be much different than for a 460 m3/s flood, especially since those values are subject to many uncertainties. Second, regarding this precise choice of **12** categories, it is based on a previous hydraulic study of the sector to establish inundation maps (Leclerc and Secretan, 2012). They produced 11 maps, for streamflow varying from 550 to 1050 m3/s and separated by an increment of 50 m3/s. We adopted this increment of 50 m3/s, but included lower streamflow values. We will add this explanation in the revised version of the manuscript.

- **p. 8, line 19-21: The content and use of the data for the 2014 flood is unclear. Please add some information.** Indeed, we will add more information. Specifically, for 2014, we know the streamflow value (and therefore the associated damage), as well as the amount spent. This allows us to calibrate $\beta_w$. This is also explained in more details in section 4.2 (p.9, lines 13-16). Unfortunately, if we can share the value of the streamflow (825 m3/s), our confidentiality agreement with the civil security prevent us to communicate the amount spent.

- **p.9, line 4-6: The basis/source of the mentioned losses is unclear. Please explain how these values were derived. In line 27, a damage curve of Leclerc et al. (2001) is mentioned. This comes too late and too vague. Explain how the curve looks like, whether it is applicable in the catchment under study or/and whether and how is was adpated to your case study.**

**HESSD**

Indeed, the damages are taken directly from Leclerc et al. (2001). It is based on a survey regarding the types of houses in the sector (1-2 stories, with/without basement...) and their value obtained from municipal evaluation. Level of submersion for different streamflow values are obtained through hydraulic simulation. Damage is deduced from this level of submersion using Gompertz law Gompertz (1825). We will add these informations in the revised version of the manuscript, but as mentioned above, we choose not to replicate the curve since we do not want to put too much emphasis on its precise values. Since it is available online, the interested reader can also easily access it.

- **p.9, line 4 and line 10: consider using "losses" instead of "damages"** We choose to use "damage" instead of "loss" in order to distinguish from the usual use of the term "loss", as in "cost-loss ratio". As described in Appendix A, under risk aversion the two are not necessarily equivalent. We prefer using "damage" representing the actual, incurred, damages.

- **p. 10, line 3 to 15: Most of this should be shifted to the dicsussion section.** We agree, we will move this part to the discussion section.

- **In general, the section 4.3 is somewhat unclear and contains too many issues for discussion. Consider to shorten it to the main point that are necessary for the model application.** This is related to your previous comment. We agree that some items should be moved to the discussion. It will be done in the revised version of the manuscript.

**Section 5**

- **p. 12, line 3: discuss how the true distribution of streamflow could be determined or whether it is possible to check the validity of the used distri-**

**bution.** This comment is similar to Reviewer 1's comment B (second paragraph of the section, page C3 of his/her review). It is actually not possible to determine the true distribution of streamflow. One can only approach it by using the available historical record. It is expected that a longer record will provide a better empirical estimate of the true streamflow distribution. However, there can also be various sources of non-stationarity affecting the observed streamflow values over time (e.g. changing the measurement apparatus, climate change, land-use change, etc) that can contradict the previous sentence. Regarding the "validity of the used distribution", in our study, we did not fit any parametric distribution. As for the validity of the available empirical distribution, to the best of our knowledge, no, there is no way that its validity could be verified with certainty. We will modify the sentence "Note that the history under consideration must... distribution of streamflow" for "Note that, *strictly speaking*, the history under consideration *should*... distribution of streamflow". Then we will add a brief discussion about (1) non-stationarity effects/limits in the availability of data and (2) the impossibility to compare with the "true" distribution of streamflow.

**Section 6**

- **p.13, line 22: What do you mean by "sharpness"? Accuracy?** Sharpness is a desired attribute of ensemble and probabilistic forecasts (e.g. Gneiting and Raftery, 2007) and does not correspond to accuracy. In a sharp forecast, all members are very close to one another. They are not necessarily accurate, though, as they could all be wrong. Deterministic forecasts are, by definition, infinitely sharp (as a Dirac function). We will add a precision on that point in the revised version of the manuscript.

- **What do you refer to when you mention "relatively rare and comparatively**

**small flood events"?** we mean that the "usual" flood events for the Montmorency River are much less dramatic than the predicted ones (looking at the upper tail of the predictive distribution).

Then, for a relatively risk-averse decision maker, but small level of immaterial damages (say $A = 0.005$ and $\psi \leq 10$), having no forecast is better than having ensemble forecasts (see Figures 8 and 9). This happens because the ensemble forecasts predict huge streamflow value, which are never realized. However, those dramatic predictions lead the decision maker to spend immense amounts of money.

You are right that our statement was imprecise. We will add more details in the revised version of the manuscript.

**Section 8**

- **p.17, line 19: typo "AND in terms..."** Thank you for pointing the typo on p. 17 line 19. It has been corrected in the revised version of the manuscript.

- **Figure 3, 4 and 10: Explain the abbreviations in the figure caption** We would really prefer to leave the abbreviation in the figures for two reasons (1) The acronyms (EnKF and CRPS) are already defined in the text, before the figures (please see our answer to your comments about the introduction) and (2) The use of acronyms in figure titles allows for those titles to remain relatively short, which in our opinion is better for ease of reading.

[Figure]

**References**

Beven, K.: Facets of uncertainty: epistemic uncertainty, non- stationarity, likelihood, hypothesis testing, and communication, Hydrological Sciences Journal, 61, 1652–1665, 2016.

Duan, Q., Sorroshian, S., and Gupta, V.: Optimal use of the SCE-UA global optimization method for calibrating watershed models, Journal of Hydrology, 158, 265–284, 1994.

Gneiting, T. and Raftery, A.: Strictly Proper Scoring Rules, Prediction, and Estimation, Journal of the American Statistical Association, 102, 359–378, 2007.

Gompertz, B.: On the Nature of the Function Expressive of the Law of Human Mortality, and on a New Mode of Determining the Value of Life Contingencies, Philosophical Transactions of the Royal Society of London, 115, 513–583, 1825.

Juston, J., Kauffeldt, A., Montano, B., Seibert, J., Beven, K., and Westerberg, I.: Smiling in the rain: Seven reasons to be positive about uncertainty in hydrological modelling, Hydrological Processes, 27, 1117–1122, 2013.

Leclerc, M. and Secretan, Y.: Reconstruction de la prise d'eau de l'Arrondissement Charlesbourg – Simulation hydrodynamique du secteur Canteloup, des Îlets, Trois-Saults de la rivière Montmorency, Tech. Rep. R1416, INRS-Eau and Laval University, Quebec, 2012.

Leclerc, M., Heniche, M., Secretan, Y., and Ouarda, T.: Travaux d'atténuation des risques de crue à l'eau libre de la rivière Montmorency dans le secteur des îlets – PHASE 2. Mise à jour de l'analyse hydrologique, idmensionnement des travuax d'atténuation et analyse de l'impact sur les risques résiduels de dommage aux résidences., Tech. Rep. R555, INRS-Eau, Quebec, 2000.

---

## Author Response (AR1)

**List of changes**

We thank the reviewers once again for their helpful comments and suggestions. Hereafter we list the modifications made to the original manuscript. We also provided a "track-changes" version of the manuscript, where you can see all the modifications.

**List of changes according the the comments by Reviewer 1**

**Major comment (A): The presentation of the economic framework is too short.**

**Additional references for the presentation of economic elements, specifically risk aversion and utility theory:**
We added more references on those topics from pear-reviewed journals in the revised version of the manuscript: Krzysztofowicz (1986), Merz et al. (2009), Shorr (1966), Cerdá Tena and Quiroga Gómez (2008), Werner (2008), Fishburn (1989) and Pope and Just (1991). In particular, Fishburn (1989) provides a retrospective on vNM utility theory with many excerpts from the original book by von Neumann and Morgenstern (1944).

Using the above references, we extended the presentation of the economic framework (section 2 and also Appendix A) in order for it to be more informative for Earth scientists.

**About the definition of risk aversion**
We extended section 2 to explain risk aversion in greater details. Specifically, lines 25-29 on page 3 of now read: "Risk aversion" refers to an attribute of a decision maker who would be willing to pay a certain amount of money to remove any risk associated to a decision problem. The specific amount of money he or she is willing to pay for this is initially unknown and can be seen as an indirect measure of the magnitude of this aversion."

**About the CARA utility function and reorganization of section 2**
Section 2 was reorganised and Figure 1 was modified so that it is based (schematically) on the CARA utility function instead of on a generic function. We also expanded Appendix B, which describe the properties of the CARA utility function. First, we refer to Figure 1 in this Appendix, as you suggested. Also, the following was added (lines 3-5 page 21):
"The value of $A$ reflects the decision maker's level of risk aversion. Specifically, the *Arrow-Pratt index of absolute risk aversion* is defined as

$$A(\mu) = \frac{-\mu''(\cdot)}{\mu'(\cdot)} \tag{1}$$

for all twice continuously differentiable function $\mu(\cdot)$. If $A(\mu) > A(\tilde{\mu})$, we say that the decision maker whose preferences are represented by $\mu$ is more risk-averse than a decision maker whose preferences are represented by $\tilde{\mu}$.
Using the parametric form: $\mu(x) = \frac{-1}{A} \exp\{-Ax\}$, we immediately see that $A(\mu) = A$. Since $A(\mu)$ is independent of $x$, we say that $\mu$ exhibits a constant absolute level of risk aversion."

This shows why an increase in $A$ is equivalent to an increase in the level of risk-aversion, and why the level of risk-aversion is independent of the wealth.

**How $\mu$ reflects the decision maker's preferences regarding uncertainty**
Section 2 now explains in greater details the link between $\mu$ and the decision maker's preferences as well as the link between concavity and risk aversion. In particular, lines 5-11 page 5 now reads:
"To see why, consider the random variable $\tilde{c}$, and its expected value $\bar{c}$.[1] Since $\bar{c}$ is not risky, a risk-averse decision maker should prefer receiving $\bar{c}$ with certainty than receiving a random draw from $\tilde{c}$. That is: $U(\bar{c}) > U(\tilde{c})$, or $\mu(\bar{c}) > \sum_{m=1}^{M} p_m \mu(c_m)$, which is the definition of concavity. Note that we can also define $C > 0$, the amount of money that the decision maker would be willing to spend to remove the risk associated with $\tilde{c}$, as follows:
* * *
[1]Note that $\bar{c}$ can be thought as a degenerated random variable, taking the value $\bar{c}$ with probability 1.

$$\mu(\bar{c} - C) = \sum_{m=1}^{M} p_m \mu(c_m) \tag{2}$$

This argument extends directly to any change in risk: any risk-averse decision maker prefers less risky distributions, in the sense of mean-preserving second order stochastic dominance (Rothschild and Stiglitz, 1970). Figure 1 also presents a graphical version of the above discussion when there are only two states of nature."

Appendix A was also modified (page 20 line 23-27):
"To see more clearly the impact of risk-aversion on the optimal decision, suppose that $\mu$ is CARA, i.e. $\mu(x) = \frac{-1}{A} \exp\{-Ax\}$, and that $b = d$. Using the formula above and straightforward algebra, we find that an action is optimal if

$$p \geq \frac{\exp\{Ac\} - 1}{\exp\{Ad\} - 1} \equiv t(A) \tag{3}$$

as opposed to $p \geq c/d$ for the cost-loss ratio. One can verify that $t(A)$ is strictly decreasing with $\lim_{A\to 0} t(A) = c/d$. Then, this implies that, as risk aversion increases, the decision maker requires lower confidence level (for the realisation of the adverse event) in order to take an action. The limiting case, when the decision maker is risk neutral, gives the cost-loss ratio."

**Major comment (B): How is the upper tail of the predictive distribution taken into account?**

**"States of the world"**
The term "state of the world" is now precisely defined at page 4 lines 27 to page 5 line 2:
"The set of states of the world represent the set of realizations of $\tilde{c}$ for which the decision maker has preferences For instance, in Cerdá Tena and Quiroga Gómez (2008), there are only two possible states of the world: "adverse weather" and "non adverse weather". [2] In the case of flood forecasting systems, even if the streamflow values are continuous, the decision maker may only distinguish between a finite set of implied damages. This point is discussed further in section 4.2 where a finite number of "damage categories" are specified."

**Missed events in the database and sufficiency of data to draw full conclusions:**
The revised version of the manuscript includes a new figure (Figure 11) instead of previous boxplot of differences $Q_{fcst} - Q_{obs}$. It displays histograms reporting the number of events observed in each class of events, for each forecasting system. See also the text at page 17 lines 17-25. This figure shows, among other things, that all forecasting systems generally overforecast. Missed events are not a big issue on the Montmorency watershed. However, we do agree with you that this is an important issue for flood forecasting in general.

We also discuss the issues related to the length of the data base in greater details. See page 13 lines 21-24:
"On the one hand, it is expected that a longer record will provide a better empirical estimate of the true streamflow distribution. On the other hand, there can also be various sources of non-stationarity affecting the observed streamflow values over time (e.g. changing the measurement apparatus, climate change, land-use change, etc). Hence, even with a very long historical record, the true distribution of streamflow cannot be known with certainty. (Note that this also affects measures of quality, such as the CRPS.)"

**Major comment (C): The discussion consider non scientific issues which some hydrologists and forecasters can disagree with.**

The discussion was revised according to the comments from both reviewers. First, a portion of text was moved from section 4.3 to the Discussion (line 27 page 18 to line 7 page 28, new version) according to a suggestion from Reviewer 2.
We will not recopy the new discussion here as it is long, but you can see the changes in the "track-changes" version of the manuscript. Those changes follow your suggestions, namely:
* * *
[2]**vNM utility functions can also account for an infinite number of states of the world. In such case, one would have: $U(\tilde{c}) = \int \mu(c) f(c) dc$, where $f$ is the pdf of $\tilde{c}$.**

- We removed any part of text that could have suggested that forecasters should modify their true belief.

- We emphasized that "In any case, it is capital to recall that the role of the forecaster is to issue the best possible streamflow forecast given their knowledge of the situation and available model and data." (page 18 lines 21-24)

- We moderated the conclusion, to avoid going to far given the case study. To do this, and limited to your comments, as expressed on page C4 of your review.

- We emphasize that in our study, we consider a well-trained decision maker, but we mention the issue of training and also potential cognitive biases. We also added references (in hydrology) in which the issue of training is discussed (Ramos et al., 2013; Demeritt et al., 2010; Doswell, 2004).

Lastly, we added more precision regarding the fact that forecasts users (and people in general) are not aware of their precise level of risk aversion. Most people have a general idea of their behaviour toward risky situations (i.e. for instance a person who likes to gamble can assume that she likes taking risks, at least to some extent), but it is hard to pinpoint the precise value of $A$. For this reason, lines 7-9 page 18 now reads:

"The "real" level of risk aversion for the decision maker for flood emergency measures along the Montmorency River remains unknown due to the insufficient record of decisions and associated spending. However, it can be reasonably assumed that they are highly risk-averse (Claude Pigeon, personal communications)."

To illustrate this further, we can mention that, on several occasions during phone calls related to this work, Mr. Pigeon worried about the potential occurrence of deaths related to flooding. We asked if there has ever been deaths caused directly or indirectly by floods on the Montmorency River. His answer was: "No, but there *could* be, someday". Of course this is a very specific example, and maybe extreme, but it is similar to the example of the Minister from Prague in the manuscript. It is probably safe to say that most decision makers involved in flood mitigation will "play on the safe side" and "not take any chance" with peoples' property (and life!). While this doesn't provide a precise value for $A$ in a utility function, this is indicates a risk-averse behaviour, which cannot be represented by the cost-loss ratio.

Appendix A was also further modified to include a simple mathematical demonstration of the impossibility of considering risk-aversion in the cost-loss ratio framework (page 20 line 23-27, already mentionned above). A similar demonstration, is also performed in Cerdá Tena and Quiroga Gómez (2008).

**Detailed comments**

We will simply refer to the line of the modified manuscript addressing each comment

**Your comments from Page 2:**

**Line 4: "uncertainty \*assessment of\* hydrological forecasts conveys important information for decision makers' rather 'uncertainty in hydrological forecasts conveys important information ..."**
Page 2 line 5

**Line 7: I agree that the analog forecasts are of common use. However, why quoting this approach first? Is it used by the DEH? How is it relevant for this article? I suggest the authors list the most importance uncertainty sources (in a sorted way) and then present the methodologies which can be used to deal with them. The link between analog forecasts and then ensemble forecasts (line 13) is not clear.**
Page 2 in general, and specifically lines 7-14 for sources of uncertainty.

**Line 13 ("ensemble forecasts are superior to deterministic ones"): do the authors focus on ensemble forecasts or is it true as well for probabilistic forecasts? (ensemble forecasts being used as "proto" or substitute of probabilistic forecasts, since probabilistic information is drawn from this kind of forecasts).**
Page2 Line 20

**Lines 16-18: I agree that economic value assessment is not straightforward. However, assessing a forecast system by comparing forecasts with corresponding observations is not straightforward either.**

Indeed, there is not one quality but different qualities (especially for probabilistic (and then ensemble) forecasts). Different end-users would give different weights to these qualities (since they have specific applications).
Page 2 Lines 28-29

**Line 26 ("which does not fully exploit the information about forecast uncertainty"): what does "fully" mean here? Verkade and Werner (2011) do take explicitly into account the uncertainty.**
Page 3, lines 1-2.

**Line 31: check spelling (Neumann / Newman)**
Page 3, line 6

**Line 32: since the proposed framework is based on the von Neumann and Morgenstern utility function, more references are needed than a single one of 1944. Another reference is given further (page 3, line 30). But it is a book, which may be a "classic" in the economic community, but not the easiest reference to find and read by a hydrologist.**
Indeed. Please see our answer to the general comment A.

**Your comments from Page 3:**

**Lines 5 and 6: this sentence provides some conclusions of the article. Why here in the introduction?**
Removed

**Line 12 ("Results are presented and discussed in section 6"): in sections 6 and 7. Line 18 ("Most importantly"): do the authors mean "More importantly"?**
Page 3, line 24

**Line 23: check English (spelling for "weighting")**
Page 4 line 8

**Line 30: see comment for page 2, line 32.**
Additional references were added.

**Your comments from Page 4:**

**Line 5 ("the curvature of the function mu reflects the decision maker's preference regarding uncertainty"): why? Some references would be gratefully welcome.**
Indeed. Please see our answer to the general comment A.

**Line 9: isn't a reference to Fig. 1 missing here?**

Figure 1 modified. It is now based on the CARA utility function.

**Line 20: check English ("teh")**
Page 6 line 7

**Line 21: check the numerotation of tables (table 3 is referred to before table 1 and 2)**
Done!

**Your comments from Page 5:**

**Lines 32-33 (and lines 1-2 page 6): I did not understand why the HYDROTEL file system is useful for the reader. Are these technical details significant for this study or may they be avoided?**
Removed (page 7)

**Your comments from Page 6:**

**Lines 31-33: I am not sure that I understood correctly. Is the meteorological forecast ensemble used here computed by the meteorological service of Canada but taken from the TIGGE dataset (for some practical reasons)? If so, it might be clearer if stated this way.**
Page 8 lines 8-9.

**Your comments from Page 7:**

**Lines 10 & 11 ("Thiboult et al. (2016) showed that the [...]"): please be more specific (for this catchment? For this area?...)**
Page 8, line 19.

**Line 16: the additive coefficients for temperature inputs and the multiplicative coefficients for precipitation inputs are huge and I assume that they are much larger than the uncertainty for these inputs. Is the whole range used in practice? Is this manual 'tuning' used for more than reducing the input uncertainty in getting a best guess? Some discussion would be useful here.**
Yes they are huge. They are the true operational limits at the DEH. However, it is worth emphasizing that the goal of those perturbations on precipitations and temperature is to (indirectly) affect state variables (soil moisture, snow water equivalent) and correct model uncertainties. They are not intended as to reflect the true uncertainty on precipitations and temperature. The goal of this manual tuning is indeed to obtain a best guess regarding the initial state of the watershed (under the assumption that the state variables of the model accurately reflect the state of the watershed). However, it might reassure the reviewer (as well as everybody else) to know that those huge limits for perturbations are rarely reached. In our study, the multiplicative coefficient applied to precipitation varied between 0.5 and 2.5. Most additive coefficients for temperature varied between -3 and +2.5, with occasional large coefficient (up to -7 and +7 on 2-3 occasions). Precisions regarding what perturbations were really applied and the limits that were permitted were added in the revised version of the manuscript.

**Page 8 lines 27 to Page 9 line 1.**
The EnKF that is implemented here follows Thiboult et al. (2016) and Mandel (2006). $M$ is the model error covariance matrix, computed before data assimilation, at each time step of the sequential data assimilation. As such, it is not updated, as it is the model's state variables that are updated according to equation (3). Of course, updating the state variables will affect the model outputs, hence M at the following time step. Thus, in our specific implementation state variables are indeed updated, but not from an open loop simulation. The base line simulation here is the manually assimilated run. This base line simulation is good, but cruelly lacks dispersion. In that context, the purpose of the EnKF is only to consider uncertainty associated to state variables and not to improve the first guess estimate of state variables. The parameters of the EnKF were not fine tuned as in many studies (such as Thiboult and Anctil, 2015), for instance. Random perturbations added to temperature were drawn from uniform distributions U[-8,+8]° and U[0.5, 1.5] (multiplicative) for precip. This choice is coherent with the way the manual data assimilation was performed, but could certainly be improved. For instance, normal error distributions are most commonly used and the spread of those distribution is calibrated until good agreement with observation is achieved. Again, the goal of the EnKF here is to add spread around best estimate of state variables, in a controlled and systematic manner. We consider that further refinement of the EnKF is outside the scope of our study.

**Your comments from Page 8:**
**Line 16: does 's' include the cost of the forecasting system (independently from the money spent for risk mitigation)?**
Page 9, lines 15-19.

**Line 21: may the author provide some figures (orders of magnitude?) or some plots?**
Unfortunately we confirmed with the Civil Security of Ville de Québec that we can't, as these are confidential.

**Line 28 ("these represent relatively small levels of risks of aversion"): may the authors provide some references?**
Page 10 line 26.

**Line 28 ("it is shown that they lead to qualitative changes in the decision makers"): here again, some**

references would help the non specialist reader.
Page 10 line 26-28
**Your comments from Page 9:**
**Line 25: as a non specialist, I was amazed by the range of the psi factor (1.5 to 10). Is this usual?**
The range of $\psi$ captures two important aspects. First, in 2014, the civil security spent around 3.5 times more than the realized material damages. This reflects the fact that (perhaps obviously) the decision maker also consider immaterial damages. Since it is extremely hard to evaluate immaterial damages, we let $\psi$ vary to (very) large values. We actually also performed simulations for values much higher than 10. They are not displayed in the current version of the manuscript as the analysis would remain the same, but the graphs would be harder to read.
We believe that $\psi = 10$ is a reasonable value. Recall that immaterial damages include any damage that cannot be easily expressed in monetary value. Those include losses in the "quality of life", avoiding law suits (including the associated bad press)... and can therefore be quite high.

**Your comments from Page 10:**

**Line 16: the accuracy of forecasts is inversely related to lead time. Is it inversely proportional to it?**
Page 11, line 31.

**Line 29: I am not sure that I understood the division of parameter $\beta_m$. Why all factors $(2, 1.75, 1.5, ...)$ are larger than 1? I would have expected weights whose sum is 1.**
Those factors reflect the benefit of early warning. They are not weights. The baseline is the 1-day ahead warning, so any early warning should be more beneficial. In practice, this reflects the fact that the population has time adjust (pack, empty their basements, arrange visits to their relatives...) before being evacuated.

**Your comments from Page 11:**

**Line 17: why 'then'? First results provided are the hydrographs (Fig. 3) on which doing the visual inspection.**
Page 13, lines 4-7.

**Your comments from Page 12:**
**Lines 24-25: I suggest that the information of Appendix C comes in the main text (it is necessary for the reader).**
Page 12 line 29 to page 13 line 1.

**Your comments from Page 13:**
**Lines 7 & 8 ("This figure shows that for 1-day forecasts, those based on meteorological ensembles and dressed deterministic forecasts have similar spread"): this is not obvious for me.**
Page 14 lines 27 to page 15 line 1.

**Line 16 ("For very short lead times, the dressed deterministic forecasts outperform those based on meteorological ensembles"): some discussion (interpretation) would be appreciated on this (common) behaviour.**
Page 15, lines 9-10.

**Line 20: in practice, how does the DEH deal with the very "jumpy" ensemble curves? Are they used by operational forecasters?**
No. The DEH doesn't use forecasts based on meteorological ensembles. They use the dressed deterministic forecasts, which are not so "jumpy". This is mentioned on page 7 (line 27-29).

**Your comments from Page 14:**
**Lines 16 & 17 ("for higher level of risk aversion [...], the decision maker SHOULD prefer the 'no forecast' situation for low levels of Psi"): doesn't the modal verb convey a notion of duty? (you are right if you do what you should do). I would rather write that the forecasting system has no (economic value) or usefulness for highly risk-averse users.**
Page 16, lines 10-11

**Your comments from Page 16:**
**Line 12 ("The economic value of a forecasting system is necessarily dependent on the level of risk aversion of the decision maker"): first, it is more the economic value of the forecasts (you have to deduced its cost to get the value of the forecasting system). Then, even if I agree on the fact that it is very common (if not always), is this "necessary"? Can it be shown?**
This sentence was removed during the rewriting of the Discussion section (page 18).

**Lines 23-26: this paragraph has to be emphasized. Morevore, communicating the forecasts in a way that the end-users would perfectly understand is a key, but it is totally different from 'overforecast'.**
Please see our answer to major comment C and also the revised Discussion (page 18)

**Your comments from Page 18**

**Appendix A: it is referred before section 2.1 but it uses the concepts presented in this section.**
Page 4, lines 7-8.

**Appendix B. Where is Fig. 1 called?**
Figure 1 was modified to avoid confusion and now represents the CARA utility function. Text has been modified accordingly. See also our answer to major comment A.

**Your comments from Pages 23 & 24:**
**Tables 1 and 2 might be merged since their comparison is highly teachingful.**
We agree and this was done. Table 1 now includes data from both those previous tables.

**Your comments from Page 25**
**Table 3 could usefully be replaced by a plot of monthly values (if data is available)**
We agree. Table 3 was removed. It is now replaced by Figure 2.

**Your comments from Page 27:**
**Fig. 1: why is not the utility function plotted for negative values? Because if c < 0, then there is no 'interest' then the utility is 0? If so, why to use it with negative values in appendix A (for example, mu(-d))?**
Figure 1 now includes negative values of $c$. Indeed, one of the advantages of working with CARA utility functions is that they are defined for any value of $c$, and not just for positive values.

**List of changes according the the comments by Reviewer 2**

Again, we will simply refer to the line of the modified manuscript addressing each comment unless it is necessary to do otherwise. More detailed replies were provided in our initial response to your comments.

**Abstract**

**- No abbreviations should be used in the abstract without explanations.**
CARA: Page 1, line 9; HEPEX: Page 2, line 2; SWE: removed; BV3C: Page 7, line 4; THORPEX: Footnote page 8.

**- A sentence summarizing the main conclusions of the work should be added.**
Page 1 line 23-24: "Hence, post-processing forecasts to avoid over-forecasting could help improving both the quality and the value of forecasts."

**Introduction**
**Again, abbreviations are not well explained. Please provide the full term when the abbreviation is used for the first time. Please check the whole paper.**
Done

**p. 2, line 7/8: What does this mean? Could you provide examples?**
According to a comment by Reviewer 1, all references to analog forecasting systems have been removed from the revised version of the manuscript.

**In general, a more structured review of the literature on uncertainty is missing. For example, different types of uncertainty (epistemic versus aleatoty/natural uncertainty) could be distinguished since they may have different effects on decisions and decision makers because epistemic uncertainty can be reduced by better data or models while aleatory uncertainty cannot. Later in the paper, this should also be discussed in the context of the study.**
More references regarding the types of uncertainty were added and discusses on page 2, lines 7-14.

**The von Neumann and Morgenstern utility function should already be briefly explained in the introduction (p. 2, line 31/32)**
Please see our answer to comment A from Reviewer 1.

**p. 3, line 2: delete "forecast" once.**
Done!

**Section 2**

**If you use a section 2.1 there should also be a section 2.2. One subheading does not make sence. Consider to delete the headline.**
The heading of previous subsection 2.1 was deleted (page 3).

**The economic model and the utility functions should be better explained. The content of the chapter referenced in line 30 (p. 3) should be briefly summarized.**
Please see our answer to major comment A by Reviewer 1. He/she also had many questions and comments regarding those topics and asked for additional references. We indicated pages and lines corresponding to modifications in our answer.

**A paragraph that bridges this section to the next should be added**
Page 5 lines 16-17.

**Starting on p. 4: Check the numbering of the equations; add numbers to all equations on p.4,9 and 12.**
Thank you for pointing this out. All equations are now numbered

**Section 3**

**Typo in line 20 (p. 4)**
This has been corrected.

**p. 4, line 28/29: consider rephrasing, check logic**
Those lines currently read: "The response time of the watershed is rapid (12 hours). The return period of damaging floods is also short. This makes emergency evacuation and flood damage a common occurrence for riverside". We would like more precisions on what to clarify. First sentence means that floods appear rapidly. Second sentence means that floods happen often. This is why it is important to have flood forecasts and an emergency plan for this particular watershed.

**In Table 1, the potential damage should be added fpr each return period.**
The revised version of the manuscript includes histograms reporting the number of events observed in each class of events, for each forecasting system. We believe that this information will be more in line with the general framework of the paper. In addition, we are worried that displaying the values derived from the flow-damage curve provided in Leclerc et al. (2000), which would be gross approximations, could lead the reader to put too much confidence in those estimates.Note also that Leclerc et al. (2000)'s report (though in French) is freely accessible on the Internet.

**p. 5, line 8/9 consider rephrasing ("cause" is used twice in this short sentence)**
Page 6, lines 17-18.

**This is unclear. The calibration performed by the DEH should be explained (as well as the meaning of DEH - see my comment on the use of abbreviations)**
Page 7, lines 7-10.

**Again, there shouldn't be a section 3.3.1 only. Please reorganize the text.**
Contrarily to our initial answer, we decided instead to add a section 3.3.2. Please see page 8.

**Section 4**

**p. 8, line 14: The use of 12 categories should be justified or better explained.**
Please see page 10, lines 2-6

**p. 8, line 19-21: The content and use of the data for the 2014 flood is unclear. Please add some information.**
Please see page 11, lines 10-13 and also page 12 lines 12-15.Unfortunately, if we can share the value of the streamflow (825 m3/s), our confidentiality agreement with the civil security prevent us to communicate the amount spent.

**p.9, line 4-6: The basis/source of the mentioned losses is unclear. Please explain how these values were derived. In line 27, a damage curve of Leclerc et al. (2001) is mentioned. This comes too late and too vague. Explain how the curve looks like, whether it is applicable in the catchment under study or/and whether and how is was adpated to your case study.**
Greater details were added on page 10 line 32 to page 11 line 3.

**p.9, line 4 and line 10: consider using "losses" instead of "damages"**
We choose to use "damage" instead of "loss" in order to distinguish from the usual use of the term "loss", as in "cost-loss ratio". As described in Appendix A, under risk aversion the two are not necessarily equivalent. We prefer using "damage" representing the actual, incurred, damages.

**p. 10, line 3 to 15: Most of this should be shifted to the dicsussion section.**
We agree, this part was moved to the discussion. Please see the bottom of page 18.

**In general, the section 4.3 is somewhat unclear and contains too many issues for discussion. Consider to shorten it to the main point that are necessary for the model application.** This is related to your previous comment. We agree that some items should be moved to the discussion. It will be done in the revised version of the manuscript. **Section 5**

**p. 12, line 3: discuss how the true distribution of streamflow could be determined or whether it is possible to check the validity of the used distribution.**

It is actually not possible to determine the true distribution of streamflow. One can only approach it by using the available historical record. We elaborate on this issue on page 13, line 19-24.

**Section 6**

**p.13, line 22: What do you mean by "sharpness"? Accuracy?**
Please see page 15, lines 16-19.
**What do you refer to when you mention "relatively rare and comparatively small flood events"?**
You are right that our statement was imprecise. We will add more details in the revised version of the manuscript.We mean that the "usual" flood events for the Montmorency River are much less dramatic than the predicted ones (looking at the upper tail of the predictive distribution).We added a precision at page 16, lines 17-20.

**Section 8**

**p.17, line 19: typo "AND in terms..."**
Corrected!

**Figure 3, 4 and 10: Explain the abbreviations in the figure caption**
We would really prefer to leave the abbreviation in the figures for two reasons (1) The acronyms (EnKF and CRPS) are already defined in the text, before the figures (please see our answer to your comments about the introduction) and (2) The use of acronyms in figure titles allows for those titles to remain relatively short, which in our opinion is better for ease of reading.

[revised manuscript text omitted]

---

## Author Response (AR2)

**Response to the second list of comments by Reviewer 1**

Once again, we would like to than Reviewer 1 for his or her very thorough review of our manuscript and for the interesting discussion that arise from his or her suggestions. We are also grateful to the editor for being so pro-active in the reviewing process.

**About the link between risk aversion and the optimal cost-loss ratio**

**One point I still not understand is the possibility of a link between 2 ideas : a) risk aversion: how much an individual is eager to pay to be insured against an event ; b) the optimal ratio of false alarms and missed events for an individual, that is the number of false alarms (which cost a little amount of money c1) in order to avoid more than 1 missed event (which costs a large amount of money c2); this question can be treated by the cost-loss ratio. Since both notions are related to the amount of money an individual is eager to pay to avoid the consequences of any (respectively: more than 1) event, it is not clear for me whether we can make a link between these 2 approaches rather than opposing them.**

Indeed, whenever the decision maker's action is binary (alarm or nothing), the optimal decision takes the form of a threshold, where an action is taken if the probability of a flood is greater than some value. In appendix A, we clearly expose such an example and show that risk-aversion effectively reduces the threshold value (action is taken for lower probabilities). However, in the context of the paper, the optimal action is not binary, and represents a level of spending. The same intuition however applies: for any given forecast, as risk-aversion increases, the decision maker is willing to spend more money.

**About the number of events in the database**

**As pointed out by the authors in the discussion and in their revised manuscript, the results of the experiment on the case study are significant only if the data is representative of the catchment behaviour**

**(see the notion of expectation, page 13). The case study here is driven over 4 years (2011-2014), thus over a (very) limited number of (significant) events. The sufficiency of data (to infer any conclusion) should be explicitly treated. It may be useful to: a) display the observations and the forecasts (at least in the supplementary materials); b) sum up all the events (e.g. discharge over a chosen threshold), and to pay a particular attention to missed events and successful forecasts: I expect that their "ratio" would strongly influence the results of this study.**

Figure 4 displays hydrographs of forecasts and observations for one year. In our opinion, this representative year is sufficient to provide an insight regarding the catchment behaviour and the model's behaviour. In addition, the revised version of the manuscript includes Figure 11, which displays the relative frequency of flood events, both forecasted and observed, over the entire period of study. Translated in terms of absolute frequencies, the observed events amount to 36 days of flooding over four years. Those include both minor and major events. This represents at least one flood per year, with various durations. In our opinion this is representative of the catchment's behaviour and sufficient to pursue the economic analysis. The following sentence was also added in the presentation of Figure 11:"Over the four year period, there has been a total number of 36 days of flooding."

**Detailed comments**

**Page 2, lines 5 - 6: if available, some reference would be useful.**

Two references were added (Ramos et al. (2013) and Demeritt et al. (2010))

**Page 2, line 14 ("the latter gain in importance"): check grammar ("gains"?)**

Thank you for pointing out this typo. It was corrected.

**Page 2, line 15 ("However, AS there exist many sources of uncertainty in hydrological processes, there also exist many means [...]"): I did not get the logical implication between these 2 ideas.**

This sentence was modified and now reads: "However, there exist multiple sources of uncertainty in hydrological processes and there also exist many means of assessing those uncertainties and building an ensemble that convey the associated information."

**Page 2 line 15 ("there also exists many means"): check grammar ("exist"?)**

Corrected!

**Page 2, line 22: since the "value" word is in italic 2 lines further, I suggest to use italic for the "quality" word as well.**

The word "quality" is now written in italics also, both on line 22 and line 24.

**Page 2, line 25: isn't a point sign missing before the words "In particular"?**

Corrected!

**Page 2, line 26 ("In particular, the usefulness of a forecast is inherently linked to the decision maker's ability to adapt their behaviour to the information provided"): I totally agree with this sentence but why is this important in the introduction? (why to state this issue here?) In a more general way, it is the ability to use the information which allows a forecast to be useful.**

We would prefer to leave this sentence as it is, in the introduction, the objective being to emphasize the difference between forecasts' quality and value.

**Page 4, line 29 ("the decision maker may only distinguish between a finite set of implied damages"). I am not sure of what the reader should understand here; is this a practical observation or a more theoretical affirmation?**

This is a practical observation. The sentence was modified and now reads: "In the case of flood forecasting systems, even if the streamflow values are continuous, **in practice** the decision maker may only distinguish between a finite set of implied damages".

**Page 5, lines 11-12: some references (such as those given in appendix B) could usefully be provided here.**

The same references provided in appendix B were added.

**Page 6, line 15: since the month of the floods of 2012 and 2014 are given below, to precise in which month the 1964 flood occurred could be useful and consistent.**

We included the month, which is November, and we also corrected a typo (the year of the historical flood is 1966 and not 1964).

**Page 6, lines 21-22 ("The greatest concern of public authorities occurs when people refuse to evacuate [...]"): this is unfortunately a too common behaviour. However, why is this information useful here to demonstrate the pertinence of the proposed economic assessment?**

We consider this to be an important example related to intangible value (or loss). It is rather difficult to assign a monetary value to the fact that people refuse to leave their house and incur risk of injuries. In the paper this is addressed using $\psi$.

**Page 6, lines 25 and following: very little is said about the spatial resolution of the model (how many RHHU? ...). It is not clear how well it is adapted to the data inputs and to the catchment (in particular, to state that it is a physics-based model).**

There are 366 RHHU for this catchment. This was added to the manuscript at line 31 on page 6. The number of RHHU is determined by "PHYSITEL", which is a GIS companion software to HYDROTEL. This number depends on many factors: the resolutions of the digital elevation map and on the available information relative to soil type and vegetation.

**Page 7, lines 3-5. The model description has been well improved (compared to the first submission). However, I still don't understand why details such as those about the vertical water budget scheme (BV3C) are relevant for this publication.**

We consider that the short description of BV3C is important for the reader to understand the general mechanics of HYDROTEL. We would prefer to keep it this way.

**Page 8, line 19 ("In a study involving 20 catchments in Quebec"). I acknowledge the fact that the interested reader can easily find Thiboult et al. (2016). However, it could be worth providing a few details about these catchments (can we compare them to the Montmorency River catchment used in this case study?).**

The catchments in Thiboult et al. (2016) vary in terms of sizes, river network densities, vegetation, topography, soil composition, etc. They certainly can't be seen as "hydrological twins" of the Montmorency catchment, but they are subject to similar hydro-meteorologic conditions. Even among themselves, they are all different. That is why the definition of "short-term", during which the uncertainty on initial conditions is dominant, varies (1 to 3 days).

The following was added on page 8: "Those catchments vary in size and other physical characteristics, but they are all subject to similar meteorological conditions, which are also shared by The Montmorency catchment."

**Page 8, line 20 ("the uncertainty for initial conditions dominate the other sources of uncertainty for short term (1-day to 3-day ahead)"): 3 days seem to be a very long period compared to the response time (12h, according to page 6, line 6). Is this correct?**

This is correct for the 20 catchments in Thiboult et al. (2016). The Montmorency catchment is smaller so it is expected that the definition of "short-term" will be different. The following was added on page 8:

"However, the Montmorency catchment has a smaller area than any of the 20 watersheds in Thiboult et al. (2016) and has a shorter response time. Consequently, the uncertainty on initial condition is expected to dominate for less than one day."

**Page 9. The description of the "rudimentary" data assimilation scheme is much clearer in this new version. However, I am still not convinced that it can be called an Ensemble Kalman filter (there is no sequential approach here). At least, the absence of a sequential scheme could usefully be pointed out.**

Our approach is indeed sequential. The Kalman gain is updated every 24h00 based on the model output and perturbed observations. The following was added in the text: "To do so, a rudimentary version of a sequential updating scheme, namely the Ensemble Kalman Filter (EnKF, Evensen, 2003) was implemented."

In addition, in the presentation of equation 4, it is now mentioned explicitly that the gain is updated sequentially.

The approach is rudimentary in the sense that no experiment was performed to optimize the distribution of the perturbations applied to the observations.

**Page 9 ("The inclusion of additive perturbations for precipitation is due to the fact that strong under-captation is suspected for this catchment."): I don't understand this. Do the authors mean "additive positive perturbations"?**

Yes, additive perturbations for precipitation were always positive (and small). As written in the manuscript, those perturbations were drawn from a uniform distribution bounded in [0, 0.5].

**Page 10, lines 1-3: the 'm' subscript is used for 2 different items: the ensemble members and the damage categories. This could infer some confusion.**

Thank you for pointing this out! The text has been modified and now reads "Strictly speaking, streamflow value associated to category $m$ $Q_m$ has a probability of occurrence $p_m$, and corresponds to a given damage $d(Q_m)$."

**Page 13, line 22 ("On the other hand, there can also be various sources of non-stationarity [...]"): this argument is rather specious. This may indeed occur but there are means to detect non-stationarity. The conclusion (lines 23-24) can not (should not?) be inferred from this argument. The representation of the expectation of the utility by its average value could be better discussed (see the main comments).**

We believe that it is important to mention those issues. Small samples are very common in hydrology, and yes, there are means to detect non-stationnarity. However, we believe that the available information in the context of hydrometeorological studies is always limited, sometimes more than others. Short samples and non-stationnarities are just examples of such limitations. The question of "truth" has been discussed by some authors, for instance by Weijs and van de Giessen (2011) in the context of forecast quality. We believe that the data available for this study has limitations, but that those limitations, typical of many hydrological studies, do not compromise the validity of the proposed method.

**Page 17, line 28 ("We find that risk-averse end-users mainly consider the less favourable scenarios"): is this really a finding of this study? If so, it should be stated "with/in this model, we find [...]".**

This has been changed accordingly. It now reads "In this paper, we..."

**Page 18, lines 23-24 ("in this paper, we did not address the issue of potential cognitive biases and training issues for end-users"): I agree and thank the authors to point out these issues. In addition, it could be worth clearly stating that risk aversion is not a cognitive bias (see the answers of the authors in the discussion, page C10), e.g. page 10 after lines 21-22.**

Indeed. We added the sentence: "However, since risk-aversion is not a cognitive bias, even highly trained decision makers are expected to be risk-averse (c.f. Fishburn (1989), Krzysztofowicz (1986))."

**Page 27, Tab. 2: I don't understand the horizontal line under "No limit for a 1-day forecast". Should other lines (above and under "No limit for a 5-day forecast") be added?**

For consistency, horizontal lines were added above and below the "No limit for a 5-day forecast".

**Page 28, Fig. 1: this figure is a "schematic" representation of the CARA utility function for A ¿ 0. Why not showing a real CARA utility function for A ¿ 0? It could help the reader keeping in mind that the CARA function values are all negatives in such a case?**

This is for practical purposes. It is easier to display a schematic representation (which has all the "visible" properties of the CARA) while being able to slightly rescale the picture so that all the relevant quantities (streamflow value, utility, expected utility) can be displayed.

**Page 28, Fig. 1 caption: is this equation coherent with the equation 2 (page 5)? Furthermore, I am also uncomfortable with this equation. Since C is the difference between 2 utility values, it is therefore not a money amount (see page 14, line 7: "the actual value of the decision maker's utility has no interpretation). Then, if the equation in the caption is correct, why and how could C be the amount that the individual would be willing to pay up to?**

You are correct. Thank you for pointing this out. The correct equation is as in equation 2 on page 5. This has been corrected.

**References**

Demeritt, D., Nobert, S., Cloke, H., and Pappenberger, F.: Challenges in communicating and using ensembles in operational flood forecasting, Meteorological Applications, 17, 209–222, 2010.

Evensen, G.: The Ensemble Kalman Filter: theoretical formulation and practical implementation, Ocean Dynamics, 53, 343–367, 2003.

Fishburn, P.: Retrospective on the Utility Theory of von Neumann and Morgenstern, Journal of Risk and Uncertainty, 2, 127–158, 1989.

Krzysztofowicz, R.: Expected utility, benefit, and loss criteria for seasonal water supply planning, Water Resources Research, 22, 303–312, 1986.

Ramos, M.-H., van Andel, S., and Pappenberger, F.: Do probabilistic forecasts lead to better decisions?, Hydrology and Earth System Sciences, 17, 2219–2232, 2013.

Thiboult, A., Anctil, F., and Boucher, M.-A.: Accounting for three sources of uncertainty in ensemble hydrological forecasting, Hydrology and Earth System Science, 20, doi:10.5194/hess-20-1809-2016, 2016.

Weijs, S. and van de Giessen, N.: Accounting for Observational Uncertainty in Forecast Verification: An Information-Theoretical View on Forecasts, Observations, and Truth, Monthly Weather Reviwe, 139, 2156–2162, 2011.